# The Price of Freedom: Exploring Tradeoffs in Equivariant Tensor Products with Spherical Signals

## Abstract

$E(3)$-equivariant neural networks have demonstrated success across a wide range of 3D modelling tasks. A fundamental operation in these networks is the tensor product, which interacts two geometric features in an equivariant manner to create new features. Due to the high computational complexity of the tensor product, significant effort has been invested to optimize the runtime of this operation. Luo et al. (2024) recently proposed the Gaunt tensor product (GTP) which promises a significant speedup over the naive implementation of the tensor product. However, this method is unable to perform antisymmetric operations which are crucial for tasks involving chirality. In this work, we introduce vector signal tensor product (VSTP) to solve this issue and show how it generalizes to a class of irrep signal tensor products (ISTPs). Finally, we investigate why these tensor products are faster. We find most of the speedup comes at the price of expressivity. Further, we microbenchmarked the various tensor products and find that the theoretical runtime guarantees may differ wildly from empirical performance, demonstrating the need for careful application-specific benchmarking. Our code is linked here.

## 1 Introduction

Many complex physical systems possess inherent spatial symmetries, and incorporating these symmetries into models has been shown to significantly improve both learning efficiency and robustness Batzner et al. (2022); Rackers et al. (2023); Frey et al. (2023); Owen et al. (2024). To address the specific symmetries present in 3D systems, considerable effort has been dedicated to the development of $E(3)$-equivariant neural networks (E(3)NNs) (Thomas et al., 2018; Weiler et al., 2018; Kondor, 2018; Kondor et al., 2018). E(3)NNs have delivered strong performance across a wide range of scientific applications, including molecular force fields (Batzner et al., 2022; Musaelian et al., 2023; Batatia et al., 2022), catalyst discovery (Liao & Smidt, 2023), generative models (Hoogeboom et al., 2022), charge density prediction (Fu et al., 2024), and protein structure prediction (Lee et al., 2022; Jumper et al., 2021).

The group $E(3)$ consists of all rotations, translations and reflections in 3 dimensions; we say a model is $E(3)$-equivariant if it satisfies:

$$f(g \cdot x) = g \cdot f(x) \quad \forall g \in E(3), x \in X \tag{1}$$

$E(3)$-equivariant neural networks work with features that transform as irreducible representations of $O(3)$, termed 'irreps', as described in Appendix A. How these irreps transform under 3D rotation, $SO(3)$ is defined by a positive integer $L$, which can intuitively be thought of as an angular frequency. To interact these irreps, a special 'tensor product' operation is performed, which replaces how features are traditionally multiplied with each other in a typical neural network. As described in Section 2, the well-studied Clebsch-Gordan (Varshalovich et al., 1988) coefficients can be used to define a tensor product. The most general Clebsch-Gordan tensor product (CGTP) has a time complexity[1] of $\mathcal{O}(L^5)$ as we show in Appendix D, which can quickly become expensive for larger $L$.

---

[1]Note that (Passaro & Zitnick, 2023) claims a runtime of $\mathcal{O}(L^6)$ for this tensor product. In Appendix D, we show that this runtime is actually $\mathcal{O}(L^5)$.

This scaling has limited the direct application of $E(3)$-equivariant neural networks to larger systems; and there is now much interest in optimizing several key operations within these neural networks.

One such optimization was identified by (Passaro & Zitnick, 2023) for the special case of when one of the inputs is derived from the spherical harmonics. Under a suitable rotation, the irreps derived from spherical harmonics become sparse, allowing for a runtime of $\mathcal{O}(L^3)$. However, the extreme reduction in sparsity is not generally true for arbitrary irrep values.

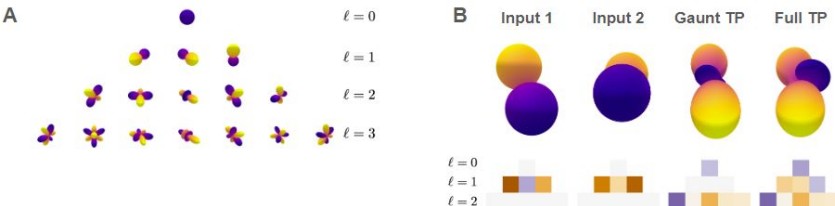

Figure 1: (A) Spherical harmonics are basis functions that transform as specific irreps of $O(3)$. (B) Gaunt Tensor Products accelerate tensor products using projections on the sphere, however this eliminates certain paths such as antisymmetric outputs.

For arbitrary irrep values, Luo et al. (2024) proposed the Gaunt Tensor Product (GTP) which they show has a complexity $\mathcal{O}(L^3)$. Further, Unke & Maennel (2024) introduced another $\mathcal{O}(L^3)$ operation which we call matrix tensor product (MTP). While this represents exciting progress, it raises an important question:

- What is fundamentally different between these new tensor products and the general CGTP with $\mathcal{O}(L^5)$ complexity?

In this paper, we show that these reductions in complexity come from merging independent interactions (what we will call paths) contained within the general tensor product. This leads to two other questions:

- How do these changes affect the expressivity of tensor product operations?
- How should we compare runtimes when models use different formulations of tensor products?

In Section 4, we give a measure of expressivity of the different tensor products and demonstrate that by normalizing against expressivity, the original GTP algorithm, MTP, and the sparse version of CGTP all have the same asymptotic runtime. Therefore the speed up come at the price of freedom in expressivity.

Based on rigorous benchmarking, GTP seems to scale better than MTP. Further, we identify that there exists asymptotically faster spherical harmonic transform algorithms (Healy et al., 2003) which allows the asymptotic runtime of GTP to be faster than the others even when normalized for expressivity. However, GTP is the only tensor product which suffers from an inability to perform anti-symmetric operations. In fact we demonstrate this prevents identification of chiral tetris pieces in Section 5.2. This motivates us to propose a new tensor product which leverages the ideas of GTP to retain the speed yet allows antisymmetry.

Therefore in Section 3.2 we propose vector signal tensor products (VSTP). Rather than using scalar signals and scalar products, VSTP uses vector signals and cross product interactions. We prove that the selection rules for VSTP only eliminate the trivial scalar $0 \otimes 0$ interaction. In addition, we show that VSTP has the same asymptotic runtime as GTP. Further, we describe in Section 3.4 how our approach generalizes to arbitrary irrep signal types to form an entire class of tensor products we call irrep signal tensor products (ISTPs).

We summarize our core contributions as follows:

- Proposal of VSTP (and generalization of ISTPs) solving the antisymmetry problem of GTP
- Systematic analysis of theoretical runtimes and expressivity of various tensor products

- Rigorous benchmarking of different tensor product implementations

We organize this paper by first introducing the existing tensor products in Section 2 and our new vector signal tensor product (VSTP) and irrep signal tensor products (ISTP) in section Section 3. We then discuss how to measure the expressivity of different tensor products and summarize the results in Section 4. Finally we benchmark the various tensor products showing that asymptotics do not always correspond with practical performance.

We assume familiarity with group representations. For a brief introduction to representations (reps) and irreducible representations (irreps), we refer the reader to Appendix A.

## 2 TENSOR PRODUCTS

We begin by motivating why tensor products between irreps are an important operation. First, irreps are a key component of many equivariant architectures Geiger & Smidt (2022); Unke & Maennel (2024). In particular, Schur's lemma states that equivariant linear maps between 2 irreps consist of either scaling if the irreps are the same, or must be 0 (Dresselhaus et al., 2007). Hence if the input and outputs of a linear layer are irreps, Schur's lemma directly gives the constraints needed for the layer to be equivariant. This is why identifying features in terms of irreps is so prevalent in equivariant architectures.

However, Schur's lemma prevents interactions between different irreps in linear layers. This is the motivation for tensor product operations. Given two spaces $V$ and $W$ and a desired output space $Z$, the most natural interaction is a bilinear mapping. This can also be viewed as a linear mapping from the tensor product space $V \otimes W \to Z$. Since we are trying to construct equivariant networks, if there are actions of group $G$ on spaces $V, W, Z$, we would like our bilinear map to be equivariant. Further, because irreps are so useful for constructing linear layers, we want the input and output reps of our bilinear maps to explicitly be written as direct sums of irreps. Note that requiring the output to explictly be written as a sum of irreps is precisely what makes tensor products expensive. In the context of this paper, a tensor product operation refers to a fixed equivariant bilinear map $V \times W \to Z$ where $V, W, Z$ have been explicitly decomposed into a direct sum of irreps.

### 2.1 CLEBSCH-GORDAN TENSOR PRODUCT

Suppose we have 2 reps in spaces $V, W$. The most natural map is $V \times W \to V \otimes W$ constructed by taking an outer product of the inputs. If the inputs are explicitly written as a direct sum of irreps, we can write the tensor product as

$$\mathbf{x} \otimes \mathbf{y} = \bigoplus_{\substack{\mathbf{x}^{(\ell_1)} \in \mathbf{x} \\ \mathbf{y}^{(\ell_2)} \in \mathbf{y}}} (\mathbf{x}^{(\ell_1)} \otimes \mathbf{y}^{(\ell_2)}) \tag{2}$$

a new basis which is the sum of tensor product reps.

The key idea of a Clebsch-Gordan tensor product is we can explicitly reduce the tensor product reps back into a direct sum of irreps with a change of basis. This change of basis is the definition of the Clebsch-Gordan coefficients, giving us

$$\mathbf{x}^{(\ell_1)} \otimes \mathbf{y}^{(\ell_2)} = \bigoplus_{\ell_3} (\mathbf{x}^{(\ell_1)} \otimes_{\mathrm{CG}} \mathbf{y}^{(\ell_2)})^{(\ell_3)} \tag{3}$$

where

$$(\mathbf{x}^{(\ell_1)} \otimes_{\mathrm{CG}} \mathbf{y}^{(\ell_2)})^{(\ell_3)}_{m_3}$$
$$= \sum_{m_1=-\ell_1}^{\ell_1} \sum_{m_2=-\ell_2}^{\ell_2} C^{\ell_3,m_3}_{\ell_1,m_1,\ell_2,m_2} \mathbf{x}^{(\ell_1)}_{m_1} \mathbf{y}^{(\ell_2)}_{m_2}. \tag{4}$$

Therefore the original tensor product can also be rewritten as a direct sum of irreps. This defines the Clebsch-Gordan tensor product (CGTP)

$$\mathbf{x} \otimes_{\mathrm{CG}} \mathbf{y} = \bigoplus_{\substack{\mathbf{x}^{(\ell_1)} \in \mathbf{x} \\ \mathbf{y}^{(\ell_2)} \in \mathbf{y}}} (\mathbf{x}^{(\ell_1)} \otimes_{\mathrm{CG}} \mathbf{y}^{(\ell_2)}). \tag{5}$$

Note that full CGTP is really just a change of basis from a sum of tensor product reps to a sum of irreps. Hence we **do not lose any information**.

The change of basis of tensor product reps into irreps is well studied. There are selection rules, which tell us the only $\ell_3$ in Equation 4 that can be non-zero Varshalovich et al. (1988).

**Proposition 2.1** (Selection rules for CGTP). *We have nonzero values of $(\mathbf{x}^{(\ell_1)} \otimes_{\mathrm{CG}} \mathbf{x}^{(\ell_2)})^{(\ell_3)}$ only if $\ell_a \leq \ell_b + \ell_c$ for all distinct choices of $a, b, c$ (equivalently $|\ell_1 - \ell_2| \leq \ell_3 \leq \ell_1 + \ell_2$).*

We will refer to any triple $[\ell_1, \ell_2, \ell_3]$ satisfying the selection rules as a valid path.

## 2.2 Gaunt Tensor Product

The Gaunt tensor product (GTP) as introduced by Luo et al. (2024) uses the intimate connection between spherical harmonics, irreps, and spherical signals. We define this relation more precisely in Appendix B. The key idea is that any rep of form $(0, 1, \ldots, L)$ can be interpreted as coefficients for the spherical harmonics and hence corresponds to a function on $S^2$.

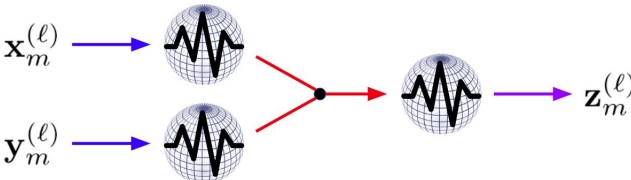

Figure 2: Schematic of the process in taking a Gaunt tensor product. We interpret input irreps as scalar SH coefficients to create spherical signals. We then take pointwise products of the two signals to create a new signal which we decompose back into scalar SH coefficients.

In particular, given two $(0, 1, \ldots, L)$ reps $\mathbf{x}$ and $\mathbf{y}$, let $f_{\mathbf{x}} = \mathrm{ToSphere}(\mathbf{x})$ and $f_{\mathbf{y}} = \mathrm{ToSphere}(\mathbf{y})$ be the associated signals on $S^2$. Taking the pointwise product of $f_{\mathbf{x}}$ and $f_{\mathbf{y}}$ on $S^2$ gives us a new function $f_{\mathbf{x}} \cdot f_{\mathbf{y}}$, also on $S^2$. Then, converting back to irreps gives us the Gaunt tensor product:

$$\mathbf{x} \otimes_{\mathrm{GTP}} \mathbf{y} = \mathrm{FromSphere}(f_{\mathbf{x}} \cdot f_{\mathbf{y}}) \tag{6}$$

Note that the outputs have only single copies of irreps. Hence GTP merges outputs of different paths such as $[\ell_1, \ell_2, \ell_2]$ and $[\ell'_1, \ell'_2, \ell_2]$ and **loses information**. In addition, note that GTP is a symmetric operation. Hence, antisymmetric paths cannot appear. For example, GTP cannot be used to compute cross products, because $\mathbf{u} \times \mathbf{v} = -\mathbf{v} \times \mathbf{u}$. We can characterize this with selection rules for GTP derived from the Gaunt coefficients (Gaunt, 1929).

**Proposition 2.2** (Selection rules for GTP). *We have nonzero values of $(\mathbf{x}^{(\ell_1)} \otimes_{\mathrm{GTP}} \mathbf{x}^{(\ell_2)})^{(\ell_3)}$ only if the following are satisfied:*

1. *$\ell_a \leq \ell_b + \ell_c$ for any distinct $a, b, c$ (equivalently $|\ell_1 - \ell_2| \leq \ell_3 \leq \ell_1 + \ell_2$)*

2. *$\ell_1 + \ell_2 + \ell_3$ is even*

Note in particular a cross product corresponds to the $[1, 1, 1]$ path which fails rule 2. In Section 5.2, we show that this implies that the Gaunt tensor product is incapable of solving a simple task of classifying chiral 3D structures.

## 2.3 Matrix Tensor Product

While this paper focuses on tensor products using spherical signals, we also mention another interaction introduced in the new `e3x` framework in the `FusedTensor` class (Unke & Maennel, 2024; Maennel et al., 2024). The main motivation for this interaction is that a tensor product rep is a matrix and we can interact 2 tensor product reps through matrix multiplication.

Hence, matrix tensor products (MTP) first takes each input and embeds the irreps in a single large enough tensor product rep using Clebsch-Gordan coefficients. After doing so, we can matrix mul-

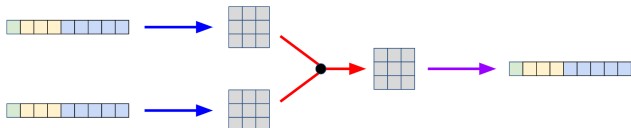

Figure 3: Schematic of the process in taking a matrix tensor product. We embed input irreps into a tensor product rep. We then interact using matrix multiplication before decomposing the resulting tensor product rep back into a direct sum of irreps.

tiply the tensor product reps. Finally, we can decompose the resulting tensor product rep back into irreps. Details are provided in Section D.3.

Similar to GTP, MTP only outputs one copy of each possible output irrep. Hence, the output irreps of the same irreps get weighted and summed together and **MTP loses information** in the same way as GTP. However in contrast to GTP, MTP is not a symmetric operation so we can have antisymmetric tensor product terms.

## 3 VECTOR SIGNAL TENSOR PRODUCT

The key idea of GTP is the connection between irreps and scalar spherical signals, which can lead to improvements in asymptotic runtime. However, the symmetry of GTP introduces an additional selection rule which eliminates antisymmetric tensor products. Here, we introduce vector signal tensor products (VSTP) which solves the antisymmetry issue while retaining the asymptotic benefits. We derive the selection rules for VSTP and prove they do not eliminate any paths except for the trivial $0 \otimes 0$. In addition, we note that in principle we can generalize GTP and VSTP to using arbitrary tensor signals which we call irreps signal tensor product (ISTP).

### 3.1 VECTOR SPHERICAL HARMONICS

Analogous to scalar spherical harmonics, one can define a set of vector spherical harmonics which forms an orthonormal basis for vector signals on a sphere.

**Definition 3.1** (Vector spherical harmonics). For integers $j, \ell, m$ where $|j - 1| \leq \ell \leq j + 1$ and $|m| \leq j$, we define the functions $\mathbf{Y}_{j,\ell}^m : S^2 \to \mathbb{R}^3$ as

$$(\mathbf{Y}_{j,\ell}^m(\hat{\mathbf{r}}))_i = \sum_{m'} \sqrt{\frac{2j+1}{2\ell+1}} C_{j,m,1,i}^{\ell,m'} Y_\ell^{m'}(\hat{\mathbf{r}}).$$

We refer to these functions as the vector spherical harmonics.

*Remark* 3.2. One may often see different definitions of vector spherical harmonics. Our definition is the one which makes the most sense from a representation theory perspective and is discussed in Appendix C.

Similar to scalar SH, vector SH also forms a complete orthonormal basis for vector signals on a sphere.

**Proposition 3.3** (Orthonormal basis). *We have*

$$\int_{S^2} \mathbf{Y}_{j,\ell}^m \cdot \mathbf{Y}_{j',\ell'}^{m'} \mathrm{d}\Omega = \delta_{mm'} \delta_{jj'} \delta_{\ell\ell'}.$$

*Further, the functions* $\mathbf{Y}_{j,\ell}^m : S^2 \to \mathbb{R}^3$ *form a complete orthonormal basis of functions* $S^2 \to \mathbb{R}^3$.

In addition, collections of our vector SH satisfy a similar equivariance property as scalar SH do.

**Proposition 3.4** (Equivariance). *The set of functions* $\mathbf{Y}_{j,\ell}^m : S^2 \to \mathbb{R}^3$ *for a given* $j, \ell$ *are equivariant. More precisely for any* $g \in SO(3)$ *we have that*

$$\sum_{m'} D_{m,m'}^j(g) \mathbf{Y}_{j,\ell}^{m'}(\hat{\mathbf{r}}) = \mathbf{D}^1(g) \mathbf{Y}_{j,\ell}^m(\mathbf{D}^1(g)\hat{\mathbf{r}}).$$

## 3.2 Vector tensor product

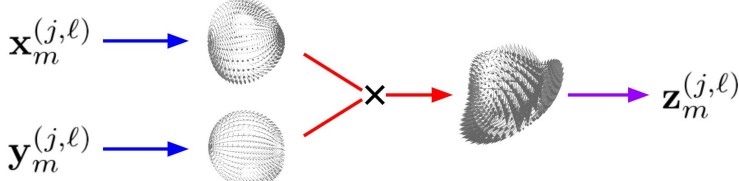

Figure 4: Schematic of the process in taking a vector signal tensor product. We interpret input irreps as vector SH coefficients to create vector spherical signals. We then take pointwise cross products of the two signals to create a new signal which we decompose back into vector SH coefficients.

Because our vector SH basis transform as irreps of type $j$, we can interpret input irreps as coefficients for these basis functions. Note however that (with the exception of $j = 1$), there are now 3 sets of basis functions which transform as an irrep of type $j$. These are the $\mathbf{Y}_{j,\ell}^m$ for $\ell = j - 1, j, j + 1$. Hence we need 3 irreps for each $j$ to specify a vector signal which we label as $\mathbf{x}^{(j,\ell)}$. Similar to the scalar case, given a $(0, 1, 1, 1, 2, 2, 2, \ldots, j, j, j)$ representation $\mathbf{x}$, we can define ToSphereVec as

$$\text{ToSphereVec}[\mathbf{x}](\hat{\mathbf{r}}) = \sum_{j,\ell,m} \mathbf{x}_m^{(j,\ell)} \mathbf{Y}_{j,\ell}^m(\hat{\mathbf{r}})$$

which converts these irreps to a vector spherical signal. Let $\mathbf{f_x}$ and $\mathbf{f_y}$ be the resulting signals from $\mathbf{x}$ and $\mathbf{y}$.

After converting the input irreps into vector signals, we can interact two signals through a pointwise cross product operation. We can then take the resulting vector signal and decompose it back into vector SH coefficients which extracts our output irreps. Hence, our output irreps are given by FromSphereVec$[\mathbf{f_x} \times \mathbf{f_y}]$.

Importantly, note that since encoding irreps into vector SH signals could be asymmetric and that pointwise cross products are antisymmetric operations, we do not automatically eliminate antisymmetric tensor products.

## 3.3 Selection rules and completeness

Similar to GTP, we can derive a set of selection rules for VSTP. For up to $j = 20$, we have tested that these selection rules are not only necessary, but also sufficient. However, proving whether these rules are sufficient is likely a hard problem as accidental zeros of the Clebsch-Gordan coefficients are still not well understood Heim et al. (2009).

**Theorem 3.5** (Selection rules for VSTP). *We have*

$$(\mathbf{x}^{(j_1,\ell_1)} \otimes \mathbf{y}^{(j_2,\ell_2)})^{(j_3,\ell_3)}$$

*is nonzero only if the following are satisfied:*

    *1. $|j_i - 1| \leq \ell_i \leq |j_i + 1|$ for $i = 1, 2, 3$*

    *2. $j_a \leq j_b + j_c$ for any distinct $a, b, c$ (equivalently $|j_1 - j_2| \leq j_3 \leq j_1 + j_2$)*

    *3. $\ell_a \leq \ell_b + \ell_c$ for any distinct $a, b, c$ (equivalently $|\ell_1 - \ell_2| \leq \ell_3 \leq \ell_1 + \ell_2$)*

    *4. $\ell_1 + \ell_2 + \ell_3$ is even*

    *5. There is no choice of distinct $a, b, c$ such that $j_a = \ell_a$ and $(j_b, \ell_b) = (j_c, \ell_c)$*

In contrast to GTP, VSTP selection rules allow all the possible paths except for multiplication of scalars.

**Theorem 3.6.** *Suppose $[j_1, j_2, j_3]$ is a valid path (are such that $|j_1 - j_2| \leq j_3 \leq j_1 + j_2$) and not all 0. Then there exists $\ell_1, \ell_2, \ell_3$ which satisfies the selection rules for VSH.*

## 3.4 GENERALIZATION: IRREP SIGNAL TENSOR PRODUCTS (ISTPs)

A natural extension is to use signals of arbitrary irrep types. We label our irrep type by $s$ in analogy to spin.

**Definition 3.7** (Tensor spherical harmonics). For integers $j, \ell, s, m$ where $|j - 1| \leq \ell \leq j + 1$ and $|m| \leq j$, we define the functions $\mathbf{Y}_{j,\ell,s}^m : S^2 \to \mathbb{R}^{2s+1}$ as

$$(\mathbf{Y}_{j,\ell,s}^m(\hat{\mathbf{r}}))_{m_s} = \sum_{m'} \sqrt{\frac{2j+1}{2\ell+1}} C_{j,m,s,m_s}^{\ell,m'} Y_\ell^{m'}(\hat{\mathbf{r}}).$$

We refer to these functions as the tensor spherical harmonics.

Similar to the scalar and vector SH, these form a complete orthonormal basis and satisfy equivariance properties. We can similarly interpret input irreps as coefficients of these basis functions and obtain tensor harmonic signals $\mathbf{f}_{\mathbf{x}_1}^{s_1} : S^2 \to \mathbb{R}^{2s_1+1}$ and $\mathbf{f}_{\mathbf{x}_2}^{s_2} : S^2 \to \mathbb{R}^{2s_2+1}$.

In general, given 2 irrep signals $\mathbf{f}_{\mathbf{x}_1}^{s_1}$ and $\mathbf{f}_{\mathbf{x}_2}^{s_2}$ we can perform pointwise CGTP operations to extract an output irrep signal of type $s_3$. This is

$$\mathbf{f}^{s_3}(\hat{\mathbf{r}}) = (\mathbf{f}_{\mathbf{x}_2}^{s_1}(\hat{\mathbf{r}}) \otimes_{\mathrm{CG}} \mathbf{f}_{\mathbf{x}_2}^{s_2}(\hat{\mathbf{r}}))^{(s_3)}.$$

Decomposing the output irrep signal into coefficients of the corresponding tensor harmonics gives the output of our operation. We call this generalization irrep signal tensor products (ISTPs) and we label them with a triple $(s_1, s_2, s_3)$ to denote the input and output signal types. In this formulation, GTP corresponds to the $(0, 0, 0)$ case and our VSTP corresponds to $(1, 1, 1)$. Note that we can also have $(1, 1, 0)$ which corresponds to taking a dot product of vector signals.

## 4 ASYMPTOTIC RUNTIMES AND EXPRESSIVITY

We would like to characterize not only how expensive, but also how expressive any particular tensor product is. To do so, we consider the construction of equivariant bilinear maps

$$B : (0 \oplus \ldots \oplus L) \times (0 \oplus \ldots \oplus L) \to (0 \oplus \ldots \oplus 2L)$$

given some existing tensor product operation $T : V \times W \to Z$. Since we assume the input and outputs of $T$ are explicitly a direct sum of irreps, it is cheap to create equivariant linear layers

$$L_{\theta_V} : 0 \oplus \ldots \oplus L \to V$$
$$L_{\theta_W} : 0 \oplus \ldots \oplus L \to W$$
$$L_{\theta_Z} : Z \to 0 \oplus \ldots \oplus 2L$$

parameterized by $\theta = (\theta_V, \theta_W, \theta_Z)$. It is not hard to see from Schur's lemma the number of parameters in $\theta_V, \theta_W, \theta_Z$ is the number of irreps of degree up to $L$ in $V, W, Z$ respectively. Using these linear maps and $T$, we get a bilinear map

$$B_{\theta(\mathbf{x},\mathbf{y}),T} = T(L_{\theta_V} \mathbf{x}, L_{\theta_W} \mathbf{y}) L_{\theta_Z}.$$

The space $\mathcal{B}_T = \{B_{\theta,T} : \forall \theta\}$ then defines all bilinear maps we can construct in this way. We can then define the dimension of $\mathcal{B}$ as a measure of the expressivity of $T$. Importantly, we note the degrees of freedom of $\theta$ is $N_{\mathrm{irreps\ in}}(T) + N_{\mathrm{irreps\ out}}(T)$ but that there is a 2-fold redundancy in the overall scaling of the map. Hence for a given tensor product $T$, we can define

$$\beta(T) = N_{\mathrm{irreps\ in}}(T) + N_{\mathrm{irreps\ out}}(T) - 2$$

which is an upper bound for our expressivity measure. However, further the number $\alpha$ of irreps in the tensor product space $(0 \otimes \ldots \otimes L) \otimes (0 \otimes \ldots \otimes L)$ is the theoretical maximum dimension possible. Note this also happens to be $N_{\mathrm{irreps\ out}}$ for CGTP so

$$\alpha = N_{\mathrm{irreps\ out}}(\mathrm{CGTP}).$$

Hence,

$$\gamma(T) = \min(\beta(T), \alpha)$$

Table 1: Asymptotic runtimes and expressivity of various tensor product implementations. Note that when normalized for expressivity, most tensor products have the same asymptotics as sparse CGTP. The only true speed up comes from a fast spherical transform algorithm by Healy et al. (2003).

| Tensor Product | # Input Irreps | # Output Irreps | Runtime | Runtime / Expressivity |
|---|---|---|---|---|
| Clebsch-Gordan (Naive) | $\mathcal{O}(L)$ | $\mathcal{O}(L^3)$ | $\mathcal{O}(L^6)$ | $\mathcal{O}(L^3)$ |
| Clebsch-Gordan (Sparse) | $\mathcal{O}(L)$ | $\mathcal{O}(L^3)$ | $\mathcal{O}(L^5)$ | $\mathcal{O}(L^2)$ |
| Gaunt (Fourier) | $\mathcal{O}(L)$ | $\mathcal{O}(L)$ | $\mathcal{O}(L^3)$ | $\mathcal{O}(L^2)$ |
| Gaunt (Grid) | $\mathcal{O}(L)$ | $\mathcal{O}(L)$ | $\mathcal{O}(L^3)$ | $\mathcal{O}(L^2)$ |
| Gaunt (S2FFT) | $\mathcal{O}(L)$ | $\mathcal{O}(L)$ | $\mathcal{O}(L^2 \log^2 L)$ | $\mathcal{O}(L \log^2 L)$ |
| Matrix (Naive) | $\mathcal{O}(L)$ | $\mathcal{O}(L)$ | $\mathcal{O}(L^4)$ | $\mathcal{O}(L^3)$ |
| Matrix (Sparse) | $\mathcal{O}(L)$ | $\mathcal{O}(L)$ | $\mathcal{O}(L^3)$ | $\mathcal{O}(L^2)$ |
| Vector Signal (Grid) | $\mathcal{O}(L)$ | $\mathcal{O}(L)$ | $\mathcal{O}(L^3)$ | $\mathcal{O}(L^2)$ |
| Vector Signal (S2FFT) | $\mathcal{O}(L)$ | $\mathcal{O}(L)$ | $\mathcal{O}(L^2 \log^2 L)$ | $\mathcal{O}(L \log^2 L)$ |
| ISTP (Naive Grid) $(s_1, s_2, s_3)$ | $\mathcal{O}((s_1 + s_2)L)$ | $\mathcal{O}(s_3 L)$ | $\mathcal{O}(\tilde{s}^2 L^2 + \tilde{s}L^3)$ | $\mathcal{O}(\tilde{s}L + L^2)$ |
| ISTP (S2FFT) $(s_1, s_2, s_3)$ | $\mathcal{O}((s_1 + s_2)L)$ | $\mathcal{O}(s_3 L)$ | $\mathcal{O}(\tilde{s}^2 L^2 + \tilde{s}L^2 \log^2 L)$ | $\mathcal{O}(\tilde{s}L + L \log^2 L)$ |

gives an upper bound for expressivity of each tensor product.

In Appendix D we analyzed the asymptotic runtimes and in Appendix E we analyzed the expressivity of implementations of the various tensor products and. The results are summarized in Table 1.

From this table, we can see that the asymptotic speedup in the faster tensor products comes from a loss of expressivity. In particular, when normalizing for our expressivity measure, the only true asymptotic speedup comes from implementations leveraging a fast algorithm for spherical harmonic transforms which we refer to as a S2FFT (Healy et al., 2003).

## 5 EXPERIMENTS

### 5.1 MICROBENCHMARKING TENSOR PRODUCTS

**Overview**. To test the analysis in Table 1, we microbenchmark some of the tensor products derived using NVIDIA's Nsight Compute profiler. We report total GPU wall time and also normalized GPU walltime according to the number of degrees of freedom defined in Section 4. We analyze the total FLOPs computed by every tensor product and notice that higher wall times don't necessarily correspond to higher FLOPs and vice-versa. We further dive into this discrepancy by reporting the peak GFLOPs/s out of all of kernels executed as part of the tensor product. This summarizes the GPU utilization achieved by every tensor product.

**Setup**. We implemented all of the tensor products in JAX Bradbury et al. (2018), including a unweighted implementation of the matrix tensor product from e3x and a more GPU-friendly implementation of Clebsch-Gordan (Sparse) 1. All of the experiments were performed on an NVIDIA A5500 with 24 GB. Our inputs are randomly generated and batch size refers to number of samples used at once. Additional evaluation details, including benchmarks for CPU and other input settings (SISO, SIMO) can be found in Appendix I.

**Walltime $\neq$ FLOPs**. The first trend we report is a discrepancy between the FLOPs computed by the tensor products and their GPU wall times with Clebsch-Gordan (Sparse) having the lowest FLOPs yet a high GPU walltime. We report low peak throughput (GFLOPs/s) despite having a more GPU-friendly implementation 1. Overall, the various gaunt tensor products and matrix tensor product are able to better saturate the GPU compared to the Clebsch-Gordan tensor products. We also notice different wall time scaling with $L_{max}$ for different tensor products, even those having similar asymptotic runtimes. For Gaunt (Fourier), our code does not leverage sparsity when transforming to a 2D Fourier basis, potentially causing the slowdown.

**Clebsch-Gordan tensor products do less compute per path.** After normalizing by $\gamma(T)$ defined in Section 4, we report the Clebsch-Gordan tensor products being the fastest both in terms of walltime and FLOPs.

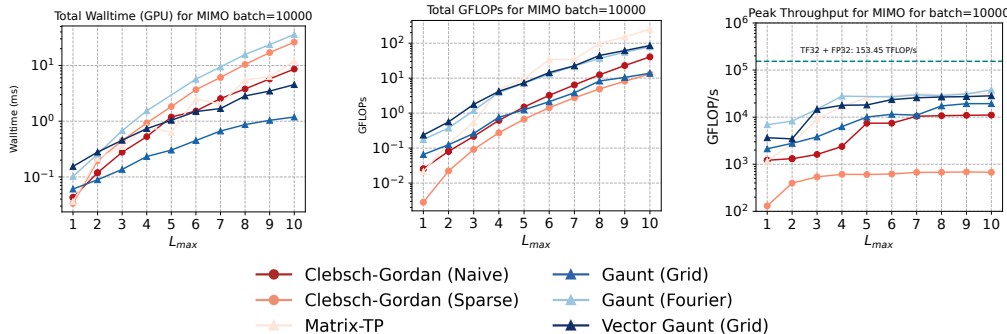

Figure 5: Analysis of tensor products compute scaling by $L_{max}$ on RTX A5500 for MIMO: (Left) Total Walltime, Total GFLOPs, and Peak Throughput (GFLOPs/s)

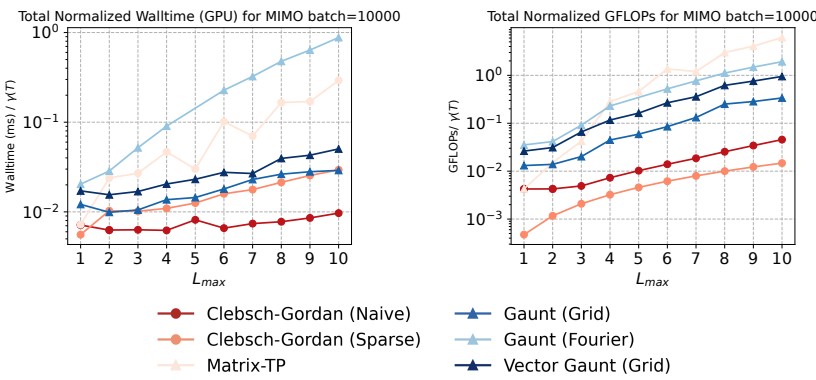

Figure 6: Analysis of tensor products compute scaling per path on RTX A5500 for MIMO (Left) Total Walltime / $\gamma(T)$ and (Right) Total GFLOPs / $\gamma(T)$

### 5.2 AN EXAMPLE WITH ANTISYMMETRY: CLASSIFYING 3D TETRIS PIECES

We consider a simple task of classifying 8 different 3D Tetris-like pieces, shown in 7a. Note that the first two pieces are non-superimposable mirror reflections of each other; they are *chiral*. Given a randomly oriented 3D structure, the network needs to predict which of the 8 tetris pieces it corresponds to.

We use a simple message-passing neural network, described in Appendix H, using either the Gaunt and Clebsch-Gordan tensor products. Our network architecture is almost identical to that of NequIP (Batzner et al., 2022).

The pieces are normalized such that the side length of each cube is 1. When represented as a graph, the center of each cube is a node. We instantiate the network with $d_{\max} = 1.1$ so that the centers are connected only to its immediately adjacent centers. The networks finally outputs $\mathbf{x} = 7 \times 0e + 1 \times 0o$ irreps. (As a reminder, $0e$ are scalars and $0o$ are pseudoscalars). The logits and predicted probabilities are then computed by:

$$l_0 = \mathbf{x}^{(0o)} \times \mathbf{x}^{(0e)_0} \qquad l_1 = -\mathbf{x}^{(0o)} \times \mathbf{x}^{(0e)_0} \qquad l_i = \mathbf{x}^{(0e)_i} \quad \text{for } i \geq 2$$

$$p_i = \text{softmax}(l_i)$$

It is clear that defining the logits in this manner preserves the rotational and reflection symmetries. The predictions are clearly invariant under rotations (as they are $\ell = 0$ irreps), and under reflections: $\mathbf{x}^{(0o)} \to -\mathbf{x}^{(0o)}$ but $\mathbf{x}^{(0e)_i} \to \mathbf{x}^{(0e)_i}$.

We set the number of message-passing steps $T$ to be 3, to allow the interactions $1o \otimes 1o \to 1e$ and then $1e \otimes 1o \to 0o$, so the pseudoscalar can be created. The degree of spherical harmonics is kept as $\ell = 4$. The irreps of the hidden layers are restricted to some cutoff $L$, which is varied from 1 to 4

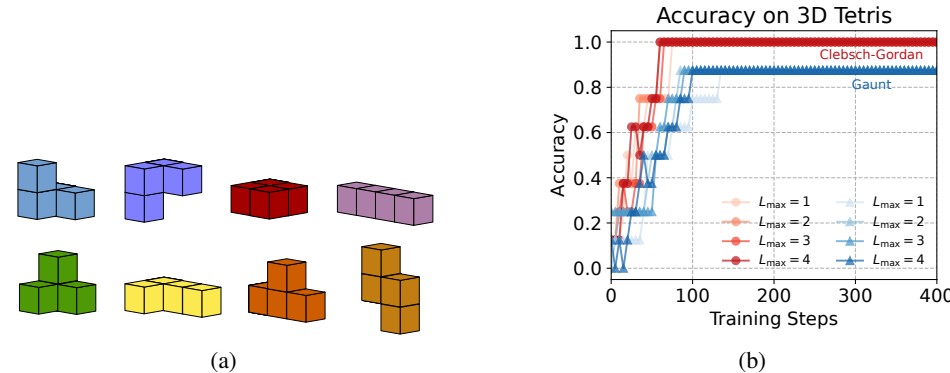

(a)                      (b)

Figure 7: (a) The 8 different 3D Tetris pieces, with the first two pieces being mirror images of each other. (b) Training curves of networks trained with different tensor products on the 3D Tetris task. The maximum $L$ is varied from $1$ to $4$. All of the Clebsch-Gordan networks attain $100\%$ accuracy while none of the Gaunt networks do.

to vary the expressivity of the network. We train the model with the Adam optimizer with learning rate $0.01$ to minimize the standard cross-entropy loss to one-hot encoded labels for the $8$ pieces.

As shown in Figure 7, the network is very easily able to solve this task with CGTP, but the same network parametrized with GTP is unable to distinguish between the two chiral pieces. Adding more channels or incorporating the pseudo-spherical harmonics (which have the opposite parity of the spherical harmonics under reflection) did not help. The fundamental failure is the inability to create the $1e$ term via $1o \otimes 1o \to 1e$ because this is the cross product, ie an antisymmetric operation. Indeed, there is no way to create a pseudoscalar using the GTP in this setting.

## 6   CONCLUSION

In this work, we investigate the distinction between different $O(3)$ equivariant tensor products that exist in the literature and analyze their asymptotic behavior, empirical runtimes, and expressivity. While there is much focus on improving how runtimes scale with $L$, we show that this speedup comes at the cost of expressivity.

This broader investigation was inspired by the observation that specific antisymmetric paths were missing in GTP and that paths which result in the same output irrep type are merged together. We identify selection rules to characterize whether certain paths are missing and formulate a measure for how merging of paths can affect expressivity. This framing lets us more easily evaluate the balance between expressivity and efficiency of new versions tensor products, which we hope others find useful. We use our framing to evaluate the asymptotic runtime and expressivity of various tensor product algorithms, including our new VSTP and the more general class of ISTPs.

Finally, we microbenchmarked different implementations of CGTP, GTP, MTP, and VSTP. As expected, the absolute walltimes of GTP, MTP, and VSTP are faster than CGTP past some $L_{max}$, but when normalized for expressivity CGTP is the fastest.

However, more empirical work is needed to understand how much the loss of expressivity affects actual performance of E(3)NNs. If we need the extra expressivity then clearly CGTP is the better choice, but if not, we may benefit from the absolute speed up of the other tensor products. Our work highlights the need to carefully analyze the tradeoffs between expressivity and walltime when deciding which tensor product operation to use in practice. There are many opportunities for creative design of equivariant operations. The best solution may depend heavily on the task and dataset at hand.

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

## A    IRREDUCIBLE REPRESENTATIONS OF $E(3)$

A representation $\rho$ of a group $G$ maps each group element $g$ to a bijective linear transformation $\rho(g) \in \mathrm{GL}(V)$, where $V$ is some vector space. Representations must preserve the group multiplication property:

$$\rho(g \cdot h) = \rho(g) \circ \rho(h) \quad \forall g, h \in G \tag{7}$$

Thus, the representation $\rho$ defines a group action on a vector space $V$. The dimension of the representation $\rho$ is simply defined as the dimension of the vector space $V$.

There may be subspaces $W \subset V$ which are left invariant under actions of $\rho(g)$ for all $g \in G$. If this is the case, then restricting to $W$ also gives a representation $\rho|_W(g) \in \mathrm{GL}(W)$. If there is no nontrivial $W$, then we say the representation $\rho$ is an irreducible representation (irrep).

To build $E(3)$-equivariant neural networks, the irreducible representations of $E(3)$ play a key role. Because $E(3)$ is not a compact group, the usual approach has been to consider irreducible representations of the group $SO(3)$ of 3D rotations, and compose them with the representation in which translations act as the identity:

$$\rho(R, T) = \rho'(R) \tag{8}$$

This is why translations are often handled in $E(3)$-equivariant neural networks by centering the system or only using relative vectors.

The 'scalar' representation $\rho_{\mathrm{scalar}}$ representation of $SO(3)$ is defined as:

$$\rho_{\mathrm{scalar}}(R) = \mathrm{id} \quad \forall R \in SO(3) \tag{9}$$

and is of dimension 1 over $V = \mathbb{R}$. Elements of $\mathbb{R}$ are unchanged by any rotation $R$. We call such elements 'scalars' to indicate that they transform under the 'scalar' representation of $SO(3)$. An example of a 'scalar' element could be mass of an object, which does not change under rotation of coordinate frames.

Let $T(R) \in \mathbb{R}^{3 \times 3}$ be the rotation matrix corresponding to a rotation $R \in SO(3)$. Then, the 'vector' representation of $SO(3)$ is defined as:

$$\rho_{\mathrm{vector}}(R) = T(R) \quad \forall R \in SO(3) \tag{10}$$

and is of dimension 3 over $V = \mathbb{R}^3$. The name arises from the way vectors in $\mathbb{R}^3$ transform under a rotation of the coordinate frame. We call such elements 'vectors' to indicate that they transform under the 'vector' representation of $SO(3)$. For example, the velocity and position of an object in a certain coordinate frame are 'vectors'.

Weyl's theorem for the Lie group $SO(3)$ states that all finite-dimensional representations of $SO(3)$ are equivalent to direct sums of irreducible representations. The irreducible representations of $SO(3)$ are indexed by an integer $\ell \geq 0$, with dimension $2\ell + 1$. $\ell = 0$ corresponds to the 'scalar' representation, while $\ell = 1$ corresponds to the 'vector' representation above. We will often use $m$, where $-\ell \leq m \leq \ell$, to index of each of the $2\ell + 1$ components.

We say that a quantity $\mathbf{x} \in \mathbb{R}^{2\ell+1}$ is a $\ell$ irrep, if it transforms as the irreducible representation ('irrep') of $SO(3)$ indexed by $\ell$. If $\mathbf{x}_1$ is a $\ell_1$ irrep and $\mathbf{x}_2$ is an $\ell_2$ irrep, we say that $(\mathbf{x}_1, \mathbf{x}_2)$ is a direct sum of $\ell_1$ and $\ell_2$ irreps, which we call a $(\ell_1, \ell_2)$ 'rep'. Weyl's theorem states that all reps are a direct sum of $\ell_i$ irreps, possibly with repeats over $\ell_i$: $\mathbf{x} = \oplus_{\ell_i} \mathbf{x}^{(\ell_i)}$. The multiplicity of an irrep in a rep is exactly the number of repeats.

An important lemma for constructing equivariant linear layer is Schur's lemma (Dresselhaus et al., 2007).

**Lemma A.1** (Schur's Lemma). *Suppose $V_1, V_2$ are irreps of a Lie group over any algebraically closed field (such as $SO(3)$). Let $\phi : V_1 \to V_2$ be an equivariant linear map.*

*Then $\phi$ is either $0$ or an isomorphism.*

*Further, if $V_1 = V_2$ then $\phi$ is a multiple of identity.*

*Finally for any two $\phi_1, \phi_2 : V_1 \to V_2$ we must have $\phi_1 = \lambda \phi_2$.*

This tells us that to construct equivariant linear layers between reps written as a direct sum of irreps, we can only have weights between input and output irreps of the same type and that those weights must be tied together so they give multiples of the identity transformation.

# B  SPHERICAL HARMONICS

The spherical harmonics are intimately connected to the representations of $SO(3)$ and play a key role in the Gaunt tensor product.

We define the spherical coordinates $(r, \theta, \varphi)$ as:

$$\begin{bmatrix} x \\ y \\ z \end{bmatrix} = \begin{bmatrix} r \sin \theta \cos \varphi \\ r \sin \theta \sin \varphi \\ r \cos \theta \end{bmatrix} \tag{11}$$

for $\theta \in [0, \pi), \varphi \in [0, 2\pi)$.

The spherical harmonics $Y_{\ell,m}$ are a set of functions $S^2 \to \mathbb{R}$ indexed by $(\ell, m)$, where again $\ell \geq 0, -\ell \leq m \leq \ell$. Here, $S^2 = \{(r, \theta, \phi) \mid r = 1\}$ denotes the unit sphere.

Indeed, as suggested by the notation, the spherical harmonics are closely related to the irreducible representations of $SO(3)$. Let $Y_\ell$ be the concatenation of all $Y_{\ell,m}$ over all $m$ for a given $\ell$:

$$Y_\ell(\theta, \phi) = \begin{bmatrix} Y_{\ell,-\ell}(\theta, \phi) \\ Y_{\ell,-\ell+1}(\theta, \phi) \\ \dots \\ Y_{\ell,\ell}(\theta, \phi) \end{bmatrix} \tag{12}$$

When we transform the inputs to $Y_\ell(\theta, \phi)$, the output transforms as a $\ell$ irrep.

The spherical harmonics satisfy orthogonality conditions:

$$\int_{S^2} Y_{\ell_1,m_1} \cdot Y_{\ell_2,m_2} \, dS^2 = \delta_{\ell_1 \ell_2} \delta_{m_1 m_2} \tag{13}$$

where:

$$\int_{S^2} f \cdot g \, dS^2 = \int_{\theta=0}^{\pi} \int_{\varphi=0}^{2\pi} f(\theta, \varphi) g(\theta, \varphi) \sin \theta \, d\theta d\varphi \tag{14}$$

The orthogonality property allows us to treat the spherical harmonics as a basis for functions on $S^2$. We can linearly combine the spherical harmonics using irreps to approximate arbitrary functions on the sphere. Given a $(0, 1, \dots, L)$ rep $\mathbf{x} = (\mathbf{x}^{(0)}, \mathbf{x}^{(1)}, \dots, \mathbf{x}^{(L)})$, we can associate the function $f_{\mathbf{x}} : S^2 \to \mathbb{R}$ as:

$$f_{\mathbf{x}}(\theta, \varphi) = \sum_{\ell=0}^{L} \sum_{m=-\ell}^{\ell} \mathbf{x}_m^{(\ell)} Y_{\ell,m}(\theta, \varphi) \tag{15}$$

The function $f_{\mathbf{x}}$ is uniquely determined by $\mathbf{x}$. In particular, by the orthogonality of the spherical harmonics (Equation 13), we can recover the $\mathbf{x}_m^{(\ell)}$ component:

$$\mathbf{x}_m^{(\ell)} = \int_{S^2} f_{\mathbf{x}} \cdot Y_{\ell,m} \, dS^2 \tag{16}$$

Thus, we can define the operations ToSphere and FromSphere:

$$\mathbf{x} \xrightarrow{\text{ToSphere}} f_{\mathbf{x}} \xrightarrow{\text{FromSphere}} \mathbf{x} \tag{17}$$

# C  TENSOR SPHERICAL HARMONICS

## C.1  INTUITION

Rather than considering scalar signals on a sphere, we can in general consider signals which transform as arbitrary representations of $SO(3)$. Since arbitrary representations are direct sums of irreps, it suffices to only consider signals which transform as irreps of $SO(3)$. Let us specify the signal irrep with an integer $s$ which we will interpret as "spin". Then we are considering functions $f : S^2 \to \mathbb{R}^{2s+1}$.

Now in general, we know that the usual scalar spherical harmonics form a complete orthonormal for scalar functions on a sphere. Hence to form an orthonormal basis for a tensor function of irrep $s$, we can use $2s+1$ copies of the spherical harmonics, one for each component of the $2s+1$ dimensional output. Hence we have a total of $(2\ell + 1) \times (2s + 1)$ coefficients for the degree $\ell$ harmonics which we can naturally form into an 2D array.

Now recall that spherical harmonics naturally correspond to $SO(3)$ irreps. In particular, we note that the coefficients of scalar spherical harmonics transform by $D^\ell$ when rotating a scalar signal. However for tensor harmonics, the output is not just a set of scalars but rather an irrep which also transforms when we rotate the signal. So, we have a transformation of $D^\ell$ along the $(2\ell + 1)$ dimensional axis and a transformation of $D^s(r)$ along the $(2s + 1)$ dimension axis. Hence, the 2D array of coefficients would transform as a $\ell \otimes s$ tensor product representation.

However, we know that all representations can be reduced into a direct sum of irreps with a change of basis. We label these irreps by $j$, drawing analogy to the quantum mechanics conventions of total angular momentum from an orbital (spherical harmonics $\ell$) and spin (signal output representation $s$) components. The new set of basis functions we label as $Y_{j,\ell,s}^{m_j,m_s}$ which can be defined in terms of the scalar spherical harmonics. We detail this correspondence next.

## C.2 Scalar spherical harmonics as equivariant functions

**Lemma C.1.** *Let $\mathbb{R}^{2\ell+1}$ be the vector space of an $SO(3)$ representation of order $\ell$. Then there is a $SO(3)$-equivariant $f : S^2 \to \mathbb{R}^{2\ell+1}$ which is unique up to scaling.*

First, the scalar spherical harmonics of degree $\ell$ can in fact be understood as equivariant functions $\mathbf{Y}_\ell : S^2 \to \mathbb{R}^{2\ell+1}$ where the output transforms as an $SO(3)$ irrep of order $\ell$. Borrowing bra-ket notation from quantum mechanics, we may label the output basis as $|\ell, m_\ell\rangle$ where $m_\ell$ ranges from $-\ell$ to $\ell$. Then we obtain

$$\mathbf{Y}_\ell(\hat{\mathbf{r}}) = \sum_{m_\ell} Y_\ell^{m_\ell}(\hat{\mathbf{r}}) |\ell, m_\ell\rangle$$

where $Y_\ell^{m_\ell}(\hat{\mathbf{r}})$ are the usual spherical harmonics. When specifying coefficients of a scalar spherical harmonic, we can interpret these as $c_\ell^{m_\ell} |\ell, m_\ell\rangle$ so that they also transform under the irrep $\ell$. We then obtain a signal by taking inner products giving

$$f(\hat{\mathbf{r}}) = \sum_{\ell,m_\ell} \langle \ell, m_\ell| \bar{c}_\ell^{m_\ell} Y_\ell^{m_\ell}(\hat{\mathbf{r}}) |\ell, m_\ell\rangle = \sum_{\ell,m_\ell} \bar{c}_\ell^{m_\ell} Y_\ell^{m_\ell}(\hat{\mathbf{r}}).$$

## C.3 Tensor harmonics as scalar harmonics

For tensor harmonics, we would similarly like to specify a set of irreps as coefficients $c_j^{m_j} |j, m_j\rangle$ which then maps onto a tensor signal through taking an inner product. Note that the output signal has irrep $s$, so rather than considering equivariant functions $S^2 \to \mathbb{R}^{2\ell+1}$ we now consider equivariant functions $f : S^2 \to \mathbb{R}^{(2j+1)\times(2s+1)}$ where the output transforms as a tensor product representation $j \otimes s$. However, we can always transform the tensor product representation into a sum of irreps $\bigoplus_{\ell=|j-s|}^{j+1} \ell$. However, by Lemma C.1 the equivariant functions from $S^2$ onto irreps are precisely the scalar spherical harmonics up to scaling. So any equivariant $f$ must be of form

$$\sum_{m_j,m_s} f^{m_j,m_s}(\hat{\mathbf{r}}) |j, m_j, s, m_s\rangle = \sum_{\ell,m_\ell} n_\ell Y_\ell^m(\hat{\mathbf{r}}) |\ell, m_\ell\rangle$$

$$= \sum_{\ell,m_\ell} \sum_{m_j,m_s} n_\ell Y_\ell^{m_\ell}(\hat{\mathbf{r}}) |j, m_j, s, m_s\rangle \langle j, m_j, s, m_s| |\ell, m_\ell\rangle$$

$$= \sum_{m_j,m_s} \sum_{\ell,m_\ell} n_\ell \langle j, m_j, s, m_s|\ell, m_\ell\rangle Y_\ell^{m_\ell}(\hat{\mathbf{r}}) |j, m_j, s, m_s\rangle$$

$$= \sum_{m_j,m_s} \sum_{\ell,m_\ell} n_\ell C_{j,m_j,s,m_s}^{\ell,m_\ell} Y_\ell^{m_\ell}(\hat{\mathbf{r}}) |j, m_j, s, m_s\rangle.$$

It is now clear that we can specify a tensor harmonic in terms of scalar spherical harmonics. In particular, we can always choose exactly one of the $n_\ell$ to be nonzero, giving a basis of form

$$\sum_{m_j,m_s} Y_{j,\ell,s}^{m_j,m_s}(\hat{\mathbf{r}}) \, |j,m_j,s,m_s\rangle = \sum_{m_j,m_s} \sum_{m_\ell} n_{j,\ell} C_{j,m_j,s,m_s}^{\ell,m_\ell} Y_\ell^{m_\ell}(\hat{\mathbf{r}}) \, |j,m_j,s,m_s\rangle .$$

By construction, these are orthogonal. All that remains is to ensure proper normalization.

We note any particular tensor harmonic is $\mathbf{Y}_{j,\ell,s}^{m_j}(\hat{\mathbf{r}})$ so we would like to evaluate

$$\int_{S^2} \langle \mathbf{Y}_{j,\ell,s}^{m_j}(\hat{\mathbf{r}}), \mathbf{Y}_{j,\ell,s}^{m_j}(\hat{\mathbf{r}}) \rangle \, \mathrm{d}A = \int_{S^2} \sum_{m_s} Y_{j,\ell,s}^{m_j,m_s}(\hat{\mathbf{r}}), Y_{j,\ell,s}^{m_j,m_s}(\hat{\mathbf{r}}) \mathrm{d}A$$

$$= \int_{S^2} \sum_{m_s} \sum_{m,m'} n_{j,\ell} C_{j,m_j,s,m_s}^{\ell,m} \bar{Y}_\ell^m(\hat{\mathbf{r}}) n_{j,\ell} C_{j,m_j,s,m_s}^{\ell,m'} Y_\ell^{m'}(\hat{\mathbf{r}}) \mathrm{d}A$$

$$= \sum_{m_s} \sum_{m,m'} n_{j,\ell}^2 C_{j,m_j,s,m_s}^{\ell,m} C_{j,m_j,s,m_s}^{\ell,m'} \int_{S^2} \sum_{m_s} \bar{Y}_\ell^m(\hat{\mathbf{r}}) Y_\ell^{m'}(\hat{\mathbf{r}}) \mathrm{d}A$$

$$= \sum_{m_s} \sum_{m,m'} n_{j,\ell}^2 C_{j,m_j,s,m_s}^{\ell,m} C_{j,m_j,s,m_s}^{\ell,m'} \delta_{m,m'}$$

$$= \sum_{m_s} \sum_{m} n_{j,\ell}^2 (C_{j,m_j,s,m_s}^{\ell,m})^2$$

$$= n_{j,\ell}^2 \frac{2\ell+1}{2j+1}$$

where we used the fact that scalar SH are orthonormal and identities of Clebsch-Gordan coefficients. Hence to have a normalization of 1, we see we can set

$$n_{j,\ell} = \sqrt{\frac{2j+1}{2\ell+1}}. \tag{18}$$

This gives the conversion formula

$$Y_{j,\ell,s}^{m_j,m_s}(\hat{\mathbf{r}}) = \sum_{m_\ell} \sqrt{\frac{2j+1}{2\ell+1}} C_{j,m_j,s,m_s}^{\ell,m_\ell} Y_\ell^{m_\ell}(\hat{\mathbf{r}}). \tag{19}$$

This gives Definition 3.7.

### C.4  TENSOR HARMONIC INTERACTION

Given two tensor valued signals $f_1 : S^2 \to \mathbb{R}^{2s_1+1}$ and $f_2 : S^2 \to \mathbb{R}^{2s_2+1}$, the most general way to interact them is through the usual tensor product giving a signal $f_3 : S^2 \to \mathbb{R}^{(2s_1+1)\times(2s_2+1)}$. Of course we can always take this tensor product output representation and decompose back into a direct sum of irreps, giving $f_3 : S^2 \to \bigoplus_{s_3=|s_1-s_2|}^{s_1+s_2} \mathbb{R}^{2s_3+1}$. We can then decompose the final output signal back into coefficients for the tensor harmonics.

### C.5  VECTOR SPHERICAL HARMONICS CONVENTIONS

The case of $s = 1$ for equation 19 corresponds to vector spherical harmonics Definition 3.1. However, one commonly sees a different convention. These are the functions

$$\mathbf{Y}_\ell^m(\hat{\mathbf{r}}) = Y_\ell^m(\hat{\mathbf{r}})\hat{\mathbf{r}}$$
$$\mathbf{\Psi}_\ell^m(\hat{\mathbf{r}}) = r\nabla Y_\ell^m(\hat{\mathbf{r}})$$
$$\mathbf{\Phi}_\ell^m(\hat{\mathbf{r}}) = \hat{\mathbf{r}} \times \nabla Y_\ell^m(\hat{\mathbf{r}}).$$

This other convention corresponds to the radial, curl-less, and divergence-less components and is commonly used in electrodynamics. A conversion between the conventions can be found in Chapter 14.3 of Brown (2007).

# D    RUNTIME ANALYSIS

Here, we provide a detailed asymptotic analysis of runtimes for different tensor products. We consider 3 different settings.

- **Single Input, Single Output (SISO)**:
  Here we are computing only one path $[\ell_1, \ell_2, \ell_3]$ where $\ell_i \in \mathcal{O}(L)$.

  $$\ell_1 \times \ell_2 \rightarrow \ell_3$$

- **Single Input, Multiple Output (SIMO)**:
  Here we fix $\ell_1, \ell_3$ but allow all possible irreps generated by the respective tensor products.

  $$\ell_1 \times \ell_2 \rightarrow Z$$

- **Multiple Input, Multiple Output (MIMO)**:
  Here we only bound the $L$ that the tensor products use but allow full capacity for the input and output irreps. In the case of CGTP, we can have an arbitrary number of copies of each irrep but we assume we only use single copies of each irrep in the input.

  $$Z \times W \rightarrow Z$$

In the SISO and SIMO settings, the asymptotic runtimes of different tensor products are directly comparable. However, in the MIMO setting, we lose expressivity in some tensor products. This is discussed more in Appendix E. Note the MIMO setting is what one would typically want to use in practice.

## D.1    CLEBSCH-GORDAN TENSOR PRODUCT

The tensor product operation is defined as:

$$\otimes_{\mathrm{CG}} \mathbf{x}^{(\ell_2)})_{m_3}^{(\ell_3)} = \sum_{m_1=-l_1}^{l_1} \sum_{m_2=-l_2}^{l_2} C_{\ell_1,m_1,\ell_2,m_2}^{(\ell_3,m_3)} \mathbf{x}_{m_1}^{(\ell_1)} \mathbf{x}_{m_2}^{(\ell_2)} \tag{20}$$

where $C$ denotes the Clebsch-Gordan (CG) coefficients which can be precomputed.

### D.1.1    NAIVE RUNTIME

Let $L = \max(\ell_1, \ell_2, \ell_3)$. From Equation 20, for each $m_3$, we would need to sum over $m_1, m_2$ which range from $-\ell_1$ to $\ell_1$ and $-\ell_2$ to $\ell_2$ respectively. Hence, we expect $\mathcal{O}(L^2)$ operations. To compute the values for all $m$ which range from $-\ell_3$ to $\ell_3$, we see that computing a single $\ell_1 \otimes \ell_2 \rightarrow \ell_3$ tensor product requires $\mathcal{O}(L^3)$ operations.

### D.1.2    OPTIMIZED RUNTIME WITH SPARSITY

However, the CG coefficients are sparse. In the complex basis for the irreps, $C_{\ell_1,m_1,\ell_2,m_2}^{(\ell_3,m)}$ is nonzero only if $m_1 + m_2 = m_3$. Transforming to the real basis for the irreps, this condition becomes $\pm m_1 \pm m_2 = m_3$. In either case for a fixed $m_1$ and $m_3$, we only ever need to sum over a constant number of $m_2$'s rather than $\mathcal{O}(L)$ of them as naively expected. Therefore an implementation taking this sparsity into account gives us a runtime of $\mathcal{O}(L^2)$. This optimization was noted in Cobb et al..

## D.2    GAUNT TENSOR PRODUCT

The Gaunt Tensor Product (GTP) is based on the decomposition of a product of spherical harmonic functions back into spherical harmonics Luo et al. (2024). In particular, suppose one of our inputs $\mathbf{x}^{(\ell_1)}$ transforms as a direct sum of irreps up to some cutoff $L$ (ie. $\ell_1$ ranges from $0, \ldots, L$). We can view these irreps as coefficients of spherical harmonics which gives a spherical signal $F_1(\theta, \varphi) = \sum_{\ell_1,m_1} \mathbf{x}_{m_1}^{(\ell_1)} Y_{\ell_1,m_1}(\theta, \varphi)$. We similarly construct $F_2(\theta, \varphi) = \sum_{\ell_2,m_2} \mathbf{x}_{m_2}^{(\ell_2)} Y_{\ell_2,m_2}(\theta, \varphi)$.

Taking the product of these spherical signals gives a new signal $F_3(\theta, \varphi) = F_1(\theta, \varphi)F_2(\theta, \varphi)$. This new signal can be decomposed into spherical harmonics which we use to define the GTP. This results in

$$F_3(\theta, \varphi) = \sum_{\ell_3, m_3} (\mathbf{x}^{(\ell_1)} \otimes_{\text{GTP}} \mathbf{x}^{(\ell_2)})^{(\ell_3)}_{m_3} Y_{\ell_3, m_3}(\theta, \varphi). \tag{21}$$

### D.2.1 2D Fourier basis

Luo et al. (2024) describe an implementation which decomposes spherical harmonics into a 2D Fourier basis in their original paper introducing GTP. This also turns out to be the same implementation in Xin et al. (2021). We describe their procedure here.

Note that for any $\ell \leq L$ we can always write the spherical harmonics in the 2D Fourier basis:

$$Y_{\ell, m}(\theta, \varphi) = \sum_{-L \leq u, v \leq L} y^{\ell, m}_{u, v} e^{i(u\theta + v\varphi)} \tag{22}$$

for some coefficients $y^{\ell, m}_{u, v}$.

Hence, any signal $\mathbf{x}^{(\ell)}_m$ can be encoded as

$$F_1(\theta, \varphi) = \sum_{\ell=0}^{L} \sum_{m=-\ell}^{\ell} \sum_{-L \leq u, v \leq L} \mathbf{x}^{(\ell)}_m y^{\ell, m}_{u, v} e^{i(u\theta + v\varphi)} = \sum_{-L \leq u, v \leq L} \left( \sum_{\ell=0}^{L} \sum_{m=-\ell}^{\ell} \mathbf{x}^{(\ell)}_m y^{\ell, m}_{u, v} \right) e^{i(u\theta + v\varphi)}. \tag{23}$$

We identify the encoding

$$\mathbf{x}_{u, v} = \sum_{\ell=0}^{L} \sum_{m=-\ell}^{\ell} \mathbf{x}^{(\ell)}_m y^{\ell, m}_{u, v}. \tag{24}$$

One can observe that the $y^{\ell, m}_{u, v}$ are sparse and only nonzero when $m = \pm v$. Therefore, finding $\mathbf{x}_{u, v}$ if we have a set of irreps is $\mathcal{O}(L)$ and it is $O(1)$ if we only want one irrep. Because there are $\mathcal{O}(L^2)$ possible values for $u, v$, encoding into the 2D Fourier is $\mathcal{O}(L^3)$ if we encode all irreps up to $L$ or $\mathcal{O}(L^2)$ if encoding a single irrep.

For 2 functions of $\theta, \varphi$ encoded using a 2D Fourier basis $\mathbf{x}^1_{u, v}, \mathbf{x}^2_{u, v}$, we can compute their product using a standard 2D FFT in $\mathcal{O}(L^2 \log L)$ time. This gives some output encoded as $\mathbf{y}_{u, v}$ where now $u, v$ range from $-2L, \ldots, 2L$ to capture all information.

Finally, we decode the resulting function in the 2D Fourier basis back into a spherical harmonic basis to extract the output irreps. Suppose $-L \leq u, v \leq L$. We can always write

$$e^{i(u\theta + v\varphi)} = F^{\perp}_{u, v}(\theta, \varphi) + \sum_{\ell=0}^{L} \sum_{m=-\ell}^{\ell} z^{\ell, m}_{u, v} Y_{\ell, m}(\theta, \varphi) \tag{25}$$

where $F^{\perp}_{u, v}(\theta, \varphi)$ is some function in the space orthogonal to that spanned by the spherical harmonics. By construction, our output signal is always in the space spanned by the spherical harmonics so the orthogonal parts cancel. Hence we can write

$$\sum_{-2L \leq u, v \leq 2L} \mathbf{y}_{u, v} e^{i(u\theta + v\varphi)} = \sum_{-2L \leq u, v \leq 2L} \mathbf{y}_{u, v} \sum_{\ell=0}^{L} \sum_{m=-\ell}^{\ell} z^{\ell, m}_{u, v} Y_{\ell, m}(\theta, \varphi) \tag{26}$$

$$= \sum_{\ell=0}^{L} \sum_{m=-\ell}^{\ell} \left( \sum_{-2L \leq u, v \leq 2L} \mathbf{y}_{u, v} z^{\ell, m}_{u, v} \right) Y_{\ell, m}(\theta, \varphi) \tag{27}$$

Hence we identify:

$$\mathbf{y}^{\ell}_m = \sum_{-2L \leq u, v \leq 2L} \mathbf{y}_{u, v} z^{\ell, m}_{u, v}. \tag{28}$$

Once again, we can note that $z^{\ell, m}_{u, v}$ must be sparse and is only nonzero when $v = \pm m$. Hence, evaluating the above takes $\mathcal{O}(L)$ time since we sum over $\mathcal{O}(L)$ values of $u$ paired with constant number of $v$'s. If we only extract one irrep, then we range over $\mathcal{O}(L)$ values of $m$ giving $\mathcal{O}(L^2)$ runtime. If we extract all irreps up to $2L$ this becomes $\mathcal{O}(L^3)$.

### D.2.2 GRID TENSOR PRODUCT

Rather than use a 2D Fourier basis, we can instead represent the signal by directly giving its value for a set of points on the sphere. Quadrature on the sphere is a well-studied topic (Beentjes, 2015; Lebedev, 1976); in general, $\mathcal{O}(L^2)$ points are needed to exactly integrate spherical harmonics upto degree $L$ (McLaren, 1963). For this section, consider a product grid on the sphere formed by the Cartesian product of two 1D grids for $\theta$ and $\varphi$ with $\mathcal{O}(L)$ points each, for a total of $\mathcal{O}(L^2)$ points.

We can write:

$$F_1(\theta_j, \varphi_k) = \sum_{\ell=0}^{L} \sum_{m=-\ell}^{\ell} \mathbf{x}_m^{(\ell)} Y_{\ell,m}(\theta_j, \varphi_k) = \sum_{\ell=0}^{L} \sum_{m=-\ell}^{\ell} \mathbf{x}_m^{(\ell)} N_{\ell,m} P_\ell^m(\cos(\theta_j)) cs_m(\varphi_k) \quad (29)$$

where $N_{\ell,m}$ is some normalization factor, $P_\ell^m$ are the associated Legendre polynomials, and

$$cs_m(\varphi) = \begin{cases} \sin(|m|\varphi) & m < 0 \\ 1 & m = 0 \\ \cos(m\varphi) & m > 0 \end{cases} . \quad (30)$$

We note that we can first evaluate

$$g_m(\theta_j) = \sum_{\ell=0}^{L} \mathbf{x}_m^{(\ell)} N_{\ell,m} P_\ell^m(\cos(\theta_j)) \quad (31)$$

where we set $P_\ell^m = 0$ if $m > \ell$. If we have a set of irreps up to $L$ then we do the summation and this takes $\mathcal{O}(L)$ time. If we only have one irrep to encode then this takes $O(1)$ time. But we also have $\mathcal{O}(L)$ values of $\theta_j$ on the grid and $\mathcal{O}(L)$ values of $m$ to evaluate. This gives $\mathcal{O}(L^3)$ runtime to encode onto the grid for irreps up to $L$ and $\mathcal{O}(L^2)$ for a single irrep. Finally evaluating

$$F_1(\theta_j, \varphi_k) = \sum_{m=-\ell}^{\ell} g_m(\theta_j) cs_m(\varphi_k) \quad (32)$$

for a set of $\varphi_k$ can be done through a FFT in $\mathcal{O}(L \log L)$ time for each $\theta_j$ giving $\mathcal{O}(L^2 \log L)$ total. Hence we see encoding onto the sphere takes $\mathcal{O}(L^3)$ time for irreps up to $L$ and $\mathcal{O}(L^2 \log L)$ time for a single irrep.

For the multiplication of signals, we just have elementwise multiplication $F_3(\theta_k, \varphi_k) = F_1(\theta_k, \varphi_k) \cdot F_2(\theta_k, \varphi_k)$. Since there are $\mathcal{O}(L^2)$ grid points this takes $\mathcal{O}(L^2)$ time.

Finally, we decode the signal back into irreps. To do so we use the fact that

$$\mathbf{f}_m^{(\ell)} = \sum_{j,k} a_j F(\theta_j, \varphi_k) Y_{\ell,m}(\theta_j, \varphi_k) \quad (33)$$

for some coefficients $a_j$. This is essentially performing numerical integration of our signal against a spherical harmonic. Once again using the factorization of the spherical harmonics we get

$$\mathbf{f}_m^{(\ell)} = \sum_j \left( \sum_k F(\theta_j, \varphi_k) cs_m(\varphi_k) \right) a_j N_{\ell,m} P_\ell^m(\cos(\theta_j)). \quad (34)$$

The inner sum in parentheses can be computed in $\mathcal{O}(L)$ time and we need to compute it for $\mathcal{O}(L^2)$ values of $\theta_j, m$ pairs giving a runtime of $\mathcal{O}(L^3)$. Of course, we note that $cs$ really is just sines and cosines so alternatively we can use FFT which takes $\mathcal{O}(L^2 \log L)$ total. Computing the outer sum takes $\mathcal{O}(L)$ since we sum over $\mathcal{O}(L)$ values of $j$. For a single irrep there are $\mathcal{O}(L)$ values of $j$ giving $\mathcal{O}(L^2)$ for the outer sum. For irreps up to $\ell$ there are $\mathcal{O}(L^2)$ pairs of $\ell, m$ giving $\mathcal{O}(L^3)$ runtime for the outer sum. In total, we see going from the grid to the coefficients takes $\mathcal{O}(L^2 \log L)$ for a single irrep and $\mathcal{O}(L^3)$ for all irreps.

However, it turns out that the associated Legendre polynomials have recurrence properties which can be exploited to make transforming a set of irreps up to $L$ to the grid and a set of irreps up to $L$ back from the grid asymptotically more efficient Healy et al. (2003). The runtime for this algorithm which we will call S2FFT is $\mathcal{O}(L^2 \log^2 L)$.

### D.3 MATRIX TENSOR PRODUCT

Here we describe and analyze the time complexity of matrix tensor product. Let $L_1, L_2$ be the max $\ell$'s of the inputs and $L_3$ be the max $\ell$ of the outputs. We pick some $\tilde{\ell} = \lceil \max(L_1, L_2, L_3)/2 \rceil$ so that $\tilde{\ell} \otimes \tilde{\ell}$ when decomposed into irreps can contain all irreps of the inputs and outputs. Note in principle we could always choose larger $\tilde{\ell}$.

In the following runtime analysis, we assume $L_1 = L_2 = L$, $\tilde{l} = L$, and $L_3 = 2L$.

#### D.3.1 NAIVE RUNTIME

The first step of MTP is to convert our input irreps into a tensor product rep using Clebsch-Gordan coefficients as

$$\mathbf{X}_{m_1,m_2}^{(\ell)} = \sum_{m_3=-\ell}^{\ell} C_{\tilde{\ell},m_1,\tilde{\ell},m_2}^{\ell_3,m_3} \mathbf{x}_{m_3}^{(\ell)} \tag{35}$$

$$\mathbf{Y}_{m_1,m_2}^{(\ell)} = \sum_{m_3=-\ell}^{\ell} C_{\tilde{\ell},m_1,\tilde{\ell},m_2}^{\ell_3,m_3} \mathbf{y}_{m_3}^{(\ell)}. \tag{36}$$

Naively we sum over $\mathcal{O}(L)$ values of $m_3$ and need to do the computation for $\mathcal{O}(L^2)$ possible pairs of $m_1, m_2$. This gives $\mathcal{O}(L^3)$ naive runtime for converting a single irrep into a tensor product rep. To do so for all irreps up to $L$ the takes $\mathcal{O}(L^4)$ time.

We can then sum over tensor product reps to create

$$\mathbf{X} = \sum_{\ell} \mathbf{X}^{(\ell)} \qquad \mathbf{Y} = \sum_{\ell} \mathbf{Y}^{(\ell)}. \tag{37}$$

There are $\mathcal{O}(L)$ matrices to sum over if we have irreps up to $L$. Summing matrices takes $\mathcal{O}(L^2)$ time since our matrices are size $\mathcal{O}(L) \times \mathcal{O}(L)$. Hence, this takes $\mathcal{O}(L^3)$ time if we have irreps up to $L$. If we have a single irrep then we do not need to do anything.

We then multiply the matrices giving $\mathbf{Z} = \mathbf{X}\mathbf{Y}$. Using the naive matrix multiplication algorithm requires $\mathcal{O}(L^3)$ runtime.

Finally we can use Clebsch-Gordan to extract individual irreps giving

$$(\mathbf{x} \otimes_{\text{FTP}} \mathbf{y})_{m_3}^{(\ell_3)} = \sum_{m_1=-\ell_1}^{\ell_1} \sum_{m_2=-\ell_2}^{\ell_2} C_{\ell_1,m_1,\ell_2,m_2}^{(\ell_3,m_3)} \mathbf{Z}_{m_1,m_2}. \tag{38}$$

Again, naively we sum over $\mathcal{O}(L^2)$ pairs of $m_1, m_2$ and need to evaluate $\mathcal{O}(L)$ values of $m_3$ for $\mathcal{O}(L^3)$ conversion for single irrep. If we want all irreps up to $2L$ then we need $\mathcal{O}(L^4)$.

#### D.3.2 OPTIMIZED RUNTIME WITH SPARSITY

Similar to the CGTP, we can take sparsity of the Clebsch-Gordan coefficients into account. We have nonzero values only if $\pm m_1 \pm m_2 = m_3$. Hence in the encoding step, for fixed $m_1, m_2$ we only need to sum over constant number of $m_3$ instead of $\mathcal{O}(L)$. This gives a reduction of $L$ in encoding to tensor product rep. Similarly in the decoding step, we see for fixed $m_3$ we only need to sum over $\mathcal{O}(L)$ pairs of $m_1, m_2$. This gives a reduction of $L$ as well in decoding back into irreps.

### D.4 IRREP SIGNAL TENSOR PRODUCTS

Suppose we want to interpret our input irreps as coefficients for irrep signals of type $s$. Then for any given irrep of type $j$, it can be coefficients of any $\mathbf{Y}_{j,\ell,s}^{m_j}$ where $|j-s| \leq \ell \leq j+s$. At most there can be up to $2s+1$ choices of $\ell$. We can flip this condition and see that we also have $|\ell-s| \leq j \leq \ell+s$ so for given $\ell$ there are only up to $2s+1$ choices of input irrep $j$ which work. Hence, if we use scalar SH up to degree $L$, we can input $O(sL)$ irreps into our signal.

Next, for encoding we can first convert the input irreps into coefficients of scalar spherical harmonics. Using the definition of our tensor harmonics, we have

$$\sum_{m_j} \mathbf{x}_{m_j}^{(j,\ell)} Y_{j,\ell,s}^{m_j,m_s}(\hat{\mathbf{r}}) = \sum_{m_j} \sum_{m_\ell} \sqrt{\frac{2j+1}{2\ell+1}} C_{j,m_j,s,m_s}^{\ell,m_\ell} \mathbf{x}_{m_j}^{(j,\ell)} Y_\ell^{m_\ell}(\hat{\mathbf{r}})$$

$$= \sum_{m_\ell} \left( \sqrt{\frac{2j+1}{2\ell+1}} \sum_{m_j} C_{j,m_j,s,m_s}^{\ell,m_\ell} \mathbf{x}_{m_j}^{(j,\ell)} \right) Y_\ell^{m_\ell}(\hat{\mathbf{r}})$$

$$= \sum_{m_\ell} A_{m_s,m_\ell}^{(j,\ell)} Y_\ell^{m_\ell}(\hat{\mathbf{r}})$$

where $A_{m_s,m_\ell}^{(j,\ell)}$ are coefficients of the scalar SH. To compute these coefficients, naively we have to perform a summation over $m_j$ taking $\mathcal{O}(j)$ for each pair of $m_\ell, m_s$ giving runtime of $\mathcal{O}(js\ell)$. However, leveraging sparsity reduces this to $\mathcal{O}(s\ell)$. Finally, we can use the same method as Section D.2.2 to convert into spherical signals for each component which takes $\mathcal{O}(\ell^2 \log \ell)$ for single components giving $\mathcal{O}(s\ell^2 \log \ell) = \mathcal{O}(sL^2 \log L)$ for all components.

If we now allow all irreps, we need to compute $\mathcal{O}(sL)$ coefficients for $\mathcal{O}(s^2 L^2)$ time. Next, we can compute

$$B_{m_s,m_\ell}^\ell = \sum_j A_{m_s,m_\ell}^{(j,\ell)}.$$

There are $\mathcal{O}(s)$ values of valid $j$ for given $\ell$ and each $A$ has $\mathcal{O}(sL)$ components for $\mathcal{O}(s^2 L^2)$ time to compute the $B$'s. Finally, for each $m_s$ we can use the $B$'s to compute the signal values as in Section D.2.2 for a runtime of $\mathcal{O}(L^3)$ for each component. Hence we have $\mathcal{O}(sL^3)$ to compute all components. This gives $\mathcal{O}(s^2 L^2 + sL^3)$ runtime for converting all input irreps into an irrep signal on a grid.

Next, recall our grid has $\mathcal{O}(L^2)$ points. At each point, we perform a CGTP operation and extract an irrep $s_3$ from $s_1 \otimes s_2$. From our analysis of CGTP, leveraging sparsity this takes $\mathcal{O}(\min(s_1 s_2, s_1 s_3, s_2 s_3)) = \mathcal{O}(s_1 s_2 s_3 / \max(s_1, s_2, s_3))$ time. We do this for $\mathcal{O}(L^2)$ points for a total runtime of $\mathcal{O}(s_1 s_2 s_3 L^2 / \max(s_1, s_2, s_3))$ for the interaction.

Finally, we decompose the resulting signal back into tensor harmonic coefficients. First, we can componentwise decompose into scalar SH coefficients. From Section D.2.2 this takes $\mathcal{O}(L^3)$ time for each component for a total of $\mathcal{O}(s_3 L^3)$. If we do this for a single $\ell$ it takes $\mathcal{O}(s_3 L^2 \log L)$. Hence, we now have some $B_{m_\ell,m_s}^\ell$. To extract the individual components, we use orthogonality of the Clebsch-Gordan coefficients. That is,

$$\sum_{m_\ell,m_s} C_{j,m_j,s,m_s}^{\ell,m_\ell} C_{j,m_j',s,m_s}^{\ell,m_\ell} = (2s+1)\delta_{m_j,m_j'}.$$

Hence we obtain

$$\mathbf{z}_{m_{j_3}}^{(j_3,\ell_3)} = \sum_{m_\ell,m_s} \frac{1}{2s_3+1} \sqrt{\frac{2\ell+1}{2j+1}} C_{j_3,m_{j_3},s_3,m_s}^{\ell,m_\ell} B_{m_\ell,m_s}^\ell.$$

Leveraging sparsity of the Clebsch-Gordan coefficients, this takes $\mathcal{O}(s_3 L)$ time to extract a single irrep. To extract all irreps, we just repeat giving $\mathcal{O}(s_3^2 L^2)$ time. Hence, decoding back takes $\mathcal{O}(s_3^2 L^2 + s_3 L^3)$ time.

Letting $\tilde{s} = \max(s_1, s_2, s_3)$ we the following table which summarizes the runtimes of the various components.

Now we can always use the asymptotically fast version of S2FFT which effectively just changes the $L^3$ terms to $L^2 \log^2 L$.

We see that when the $s$'s are fixed constants, the runtimes correspond exactly to those of GTP. Hence our VSTP with $s_1 = s_2 = s_3$ has the same asymptotic runtime as GTP. We also note that at large $L$, MIMO scales with $\tilde{s}$. However, we can also use $\mathcal{O}(\tilde{s})$ more irreps so in this limit, the additional cost is balanced by performing more tensor products. However, if $L$ is small, we see that runtime

Table 2: Asymptotic runtimes of ISTPs assuming a naive grid implementation.

| | SISO | SIMO | MIMO |
|---|---|---|---|
| Encode | $\mathcal{O}((s_1+s_2)L^2\log L)$ | $\mathcal{O}((s_1+s_2)L^2\log L)$ | $\mathcal{O}((s_1^2+s_2^2)L^2+(s_1+s_2)L^3)$ |
| Interact | $\mathcal{O}(s_1s_2s_3L^2/\tilde{s})$ | $\mathcal{O}(s_1s_2s_3L^2/\tilde{s})$ | $\mathcal{O}(s_1s_2s_3L^2/\tilde{s})$ |
| Decode | $\mathcal{O}(s_3L^2\log L)$ | $\mathcal{O}(s_3^2L^2+s_3L^3)$ | $\mathcal{O}(s_3^2L^2+s_3L^3)$ |
| Total | $\mathcal{O}(s_1s_2s_3L^2/\tilde{s}+\tilde{s}L^2\log L)$ | $\mathcal{O}((s_1+s_2)L^2\log L+s_3^2L^2+s_3L^3)$ | $\mathcal{O}(\tilde{s}^2L^2+\tilde{s}L^3)$ |

Table 3: Asymptotic runtimes of ISTPs assuming asymptotically fast S2FFT grid implementation.

| | SISO | SIMO | MIMO |
|---|---|---|---|
| Encode | $\mathcal{O}((s_1+s_2)L^2\log L)$ | $\mathcal{O}((s_1+s_2)L^2\log L)$ | $\mathcal{O}((s_1^2+s_2^2)L^2+(s_1+s_2)L^2\log^2 L)$ |
| Interact | $\mathcal{O}(s_1s_2s_3L^2/\tilde{s})$ | $\mathcal{O}(s_1s_2s_3L^2/\tilde{s})$ | $\mathcal{O}(s_1s_2s_3L^2/\tilde{s})$ |
| Decode | $\mathcal{O}(s_3L^2\log L)$ | $\mathcal{O}(s_3^2L^2+s_3L^2\log^2 L)$ | $\mathcal{O}(s_3^2L^2+s_3L^2\log^2 L)$ |
| Total | $\mathcal{O}(s_1s_2s_3L^2/\tilde{s}+\tilde{s}L^2\log L)$ | $\mathcal{O}((s_1+s_2)L^2\log L+s_3^2L^2+s_3L^2\log^2 L)$ | $\mathcal{O}(\tilde{s}^2L^2+\tilde{s}L^2\log^2 L)$ |

scales as $\mathcal{O}(\tilde{s}^2L^2)$ as we increase $\tilde{s}$. Hence, it still makes sense to minimize the $\tilde{s}$ we use. Note that GTP has $\tilde{s}=0$ but prevents antisymmetric tensor products, while VSTP has $\tilde{s}=1$ and the selection rules do not prevent any tensor product paths except the trivial $0\otimes 0$. Therefore VSTP with $\tilde{s}=1$ should make the most sense in practice.

### D.5 ASYMPTOTIC RUNTIMES IN DIFFERENT SETTINGS

Table 4: Asymptotic runtimes of various tensor products for different output settings. The best performing tensor products for each output settings are highlighted in green. In the MIMO setting, the Clebsch-Gordan tensor products are highlighted in red to indicate that they can output irreps with multiplicity $> 1$, unlike the Gaunt tensor products.

| Tensor Product | SISO | SIMO | MIMO |
|---|---|---|---|
| Clebsch-Gordan (Naive) | $\mathcal{O}(L^3)$ | $\mathcal{O}(L^4)$ | $\mathcal{O}(L^6)$ |
| Clebsch-Gordan (Sparse) | $\mathcal{O}(L^2)$ | $\mathcal{O}(L^3)$ | $\mathcal{O}(L^5)$ |
| Gaunt (Original) | $\mathcal{O}(L^2\log L)$ | $\mathcal{O}(L^3)$ | $\mathcal{O}(L^3)$ |
| Gaunt (Naive Grid) | $\mathcal{O}(L^2\log L)$ | $\mathcal{O}(L^3)$ | $\mathcal{O}(L^3)$ |
| Gaunt (S2FFT Grid) | $\mathcal{O}(L^2\log L)$ | $\mathcal{O}(L^2\log^2 L)$ | $\mathcal{O}(L^2\log^2 L)$ |
| Vector Signal (Naive Grid) | $\mathcal{O}(L^2\log L)$ | $\mathcal{O}(L^3)$ | $\mathcal{O}(L^3)$ |
| Vector Signal (S2FFT) | $\mathcal{O}(L^2\log L)$ | $\mathcal{O}(L^2\log^2 L)$ | $\mathcal{O}(L^2\log^2 L)$ |
| ISTP (Naive grid) $(s_1,s_2,s_3)$ | $\mathcal{O}(s_1s_2s_3L^2/\tilde{s}+\tilde{s}L^2\log L)$ | $\mathcal{O}((s_1+s_2)L^2\log L+s_3^2L^2+s_3L^3)$ | $\mathcal{O}(\tilde{s}^2L^2+\tilde{s}L^3)$ |
| ISTP (S2FFT) $(s_1,s_2,s_3)$ | $\mathcal{O}(s_1s_2s_3L^2/\tilde{s}+\tilde{s}L^2\log L)$ | $\mathcal{O}((s_1+s_2)L^2\log L+s_3^2L^2+s_3L^2\log^2 L)$ | $\mathcal{O}(\tilde{s}^2L^2+\tilde{s}L^2\log^2 L)$ |

## E EXPRESSIVITY

Here, we analyze the expressivity, as defined in Section 4, of the various tensor products in the MIMO setting. As a reminder, we can use a tensor product to construct bilinear maps

$$B : (0 \oplus \ldots \oplus L) \times (0 \oplus \ldots \oplus L) \to (0 \oplus \ldots \oplus 2L)$$

by inserting equivariant linear layers before and after the tensor product. By Schur's lemma, the total number of input and output irreps gives the degrees of freedom for paramterizing the linear layers. There is an additional 2-fold redundancy in overall scaling so `#Input irreps` + `#Ouput irreps` $-2$ gives an upper bound on expressivity.

### E.1 CLEBSCH-GORDAN TENSOR PRODUCT

In the case of CGTP, we assume input which is a single copy of each irrep up to order $L$ for $\mathcal{O}(L)$ irreps in the input. In general, tensor products of single pairs of irreps gives $\mathcal{O}(L)$ output irreps. There are $\mathcal{O}(L^2)$ pairs for a total of $\mathcal{O}(L^3)$ output irreps.

### E.2 Gaunt tensor product

In the case of GTP, we note that coefficients of spherical harmonics corresponds to single copies of each irrep. Hence, we have $\mathcal{O}(L)$ input irreps. Similarly, in the output there is only one copy of each irrep we obtain from the spherical harmonic coefficients. By selection rules, the highest order harmonic we could obtain is of order $2L$. Hence the number of output irreps is also $\mathcal{O}(L)$.

### E.3 Matrix tensor product

In the case of FTP, we encode single copies of irreps into a tensor product rep $(L/2) \otimes (L/2)$. Hence there are $\mathcal{O}(L)$ inputs. We then perform matrix multiplication which results into a $(L/2) \otimes (L/2)$ tensor product rep. But this decomposes into $0 \oplus \ldots \oplus L$ giving $\mathcal{O}(L)$ output irreps.

### E.4 Irrep signals tensor product

In the case of ISTP $(s_1, s_2, s_3)$, we note that spherical signals to vector spaces of $\mathbb{R}^{2s+1}$ for a given spin $s$ can be thought of as $2s + 1$ copies of scalar spherical harmonics. In fact, it turns out we need $\mathcal{O}(sL)$ irreps to specify a irrep signal of spin $s$. Hence, the number of input irreps is $\mathcal{O}((s_1 + s_2)L)$ and number of resulting output irreps is $\mathcal{O}(s_3 L)$.

Since VSTP corresponds to a constant choice of $s_1, s_2, s_3$, it ends up having the same asymptotic number of input and output irreps as for GTP.

## F CGTP Sparse Algorithm

While leveraging sparsity of the Clebsch-Gordan coefficients will improve asymptotic runtime, in practice we would like an implementation which is GPU friendly. Here we present an algorithm which uses the sparsity to create a constant number of generalized convolution operations.

## G Simulating the Fully-Connected Clebsch-Gordan Tensor Product with Gaunt Tensor Products

One way to increase the expressivity of GTP is to first reweight the inputs $\mathbf{x}, \mathbf{y}$. That is, we first create

$$\mathbf{x}'^{(\ell)} = a_\ell \mathbf{x}^{(\ell)} \tag{39}$$

$$\mathbf{y}'^{(\ell)} = b_\ell \mathbf{y}^{(\ell)}. \tag{40}$$

where $a_\ell$ and $b_\ell$ are learnable weights. We then perform GTP after this reweighting and extract some output irrep(s) $\ell_3$. That is we get

$$(\mathbf{x}' \otimes_{\text{GTP}} \mathbf{y}')^{(\ell_3)}. \tag{41}$$

The analogous operation is fully connected CGTP. There may be multiple pairs of irreps which give a $\ell_3$ output. We can always weight and sum these to get

$$\sum_{\ell, \ell'} w_{\ell, \ell'} (\mathbf{x}^{(\ell)} \otimes_{\text{CG}} \mathbf{y}^{(\ell')})^{(\ell_3)} \tag{42}$$

where $w_{\ell, \ell'}$ are learnable weights.

However, even if we only care about symmetric tensor products, the weighted GTP operation is strictly less expressive than fully connected CGTP.

More concretely, suppose we have nontrivial $\ell = 2$ and $\ell = 4$ data in our inputs. From CGTP and the selection rules we see that

$$(\mathbf{x}^{(2)} \otimes_{\text{CG}} \mathbf{y}^{(2)})^{(2)} \qquad (\mathbf{x}^{(2)} \otimes_{\text{CG}} \mathbf{y}^{(4)})^{(2)} \tag{43}$$

$$(\mathbf{x}^{(4)} \otimes_{\text{CG}} \mathbf{y}^{(2)})^{(2)} \qquad (\mathbf{x}^{(4)} \otimes_{\text{CG}} \mathbf{y}^{(4)})^{(2)} \tag{44}$$

---

**Algorithm 1** CGTP sparse

---

**Require:** Irrep 1 $\mathbf{x}^{(\ell_1)}$, Irrep 2 $\mathbf{y}^{(\ell_2)}$, Clebsch-Gordan coefficients $C^{\ell_3,m_3}_{\ell_1,m_1,\ell_2,m_2}$

    **for** $m_3 = -\ell_3, \ldots, \ell_3$ **do**:

        **for** $m_1 = -\ell_1, \ldots, \ell_1$ **do**:

            $A^{\ell_3,m_3}_{\ell_1,m_1,\ell_2} \leftarrow C^{\ell_3,m_3}_{\ell_1,m_1,\ell_2,m_1+m_3}$

            $C^{\ell_3,m_3}_{\ell_1,m_1,\ell_2,m_1+m_3} \leftarrow 0$

    **for** $m_3 = -\ell_3, \ldots, \ell_3$ **do**:

        **for** $m_1 = -\ell_1, \ldots, \ell_1$ **do**:

            $B^{\ell_3,m_3}_{\ell_1,m_1,\ell_2} \leftarrow C^{\ell_3,m_3}_{\ell_1,m_1,\ell_2,m_1-m_3}$

            $C^{\ell_3,m_3}_{\ell_1,m_1,\ell_2,m_1-m_3} \leftarrow 0$

    **for** $m_3 = -\ell_3, \ldots, \ell_3$ **do**:

        **for** $m_1 = -\ell_1, \ldots, \ell_1$ **do**:

            $C^{\ell_3,m_3}_{\ell_1,m_1,\ell_2} \leftarrow C^{\ell_3,m_3}_{\ell_1,m_1,\ell_2,-m_1+m_3}$

            $C^{\ell_3,m_3}_{\ell_1,m_1,\ell_2,-m_1+m_3} \leftarrow 0$

    **for** $m_3 = -\ell_3, \ldots, \ell_3$ **do**:

        **for** $m_1 = -\ell_1, \ldots, \ell_1$ **do**:

            $D^{\ell_3,m_3}_{\ell_1,m_1,\ell_2} \leftarrow C^{\ell_3,m_3}_{\ell_1,m_1,\ell_2,-m_1-m_3}$

            $C^{\ell_3,m_3}_{\ell_1,m_1,\ell_2,-m_1-m_3} \leftarrow 0$

    **for** $m_3 = -\ell_3, \ldots, \ell_3$ **do**

        **for** $m_1 = -\ell_1, \ldots, \ell_1$ **do**

            $\mathbf{z}^{(\ell_3)}_{m_3} \leftarrow \mathbf{z}^{(\ell_3)}_{m_3} + A^{\ell_3,m_3}_{\ell_1,m_1,\ell_2}\mathbf{x}^{(\ell_1)}_{m_1}\mathbf{y}^{(\ell_2)}_{m_1+m_3}$

    **for** $m_3 = -\ell_3, \ldots, \ell_3$ **do**

        **for** $m_1 = -\ell_1, \ldots, \ell_1$ **do**

            $\mathbf{z}^{(\ell_3)}_{m_3} \leftarrow \mathbf{z}^{(\ell_3)}_{m_3} + B^{\ell_3,m_3}_{\ell_1,m_1,\ell_2}\mathbf{x}^{(\ell_1)}_{m_1}\mathbf{y}^{(\ell_2)}_{m_1-m_3}$

    **for** $m_3 = -\ell_3, \ldots, \ell_3$ **do**

        **for** $m_1 = -\ell_1, \ldots, \ell_1$ **do**

            $\mathbf{z}^{(\ell_3)}_{m_3} \leftarrow \mathbf{z}^{(\ell_3)}_{m_3} + C^{\ell_3,m_3}_{\ell_1,m_1,\ell_2}\mathbf{x}^{(\ell_1)}_{m_1}\mathbf{y}^{(\ell_2)}_{-m_1+m_3}$

    **for** $m_3 = -\ell_3, \ldots, \ell_3$ **do**

        **for** $m_1 = -\ell_1, \ldots, \ell_1$ **do**

            $\mathbf{z}^{(\ell_3)}_{m_3} \leftarrow \mathbf{z}^{(\ell_3)}_{m_3} + D^{\ell_3,m_3}_{\ell_1,m_1,\ell_2}\mathbf{x}^{(\ell_1)}_{m_1}\mathbf{y}^{(\ell_2)}_{-m_1-m_3}$

    **return** $\mathbf{z}^{(\ell_3)}$

---

are all nonzero. In particular, it is possible to create a $\ell = 2$ output of

$$(\mathbf{x}^{(2)} \otimes_{\mathrm{CG}} \mathbf{y}^{(2)})^{(2)} + (\mathbf{x}^{(4)} \otimes_{\mathrm{CG}} \mathbf{y}^{(4)})^{(2)}$$

with a fully connected CGTP. However, GTP instead gives a single $\ell = 2$ output of form

$$c^2_{2,2}(\mathbf{x}'^{(2)} \otimes_{\mathrm{CG}} \mathbf{y}'^{(2)})^{(2)} + c^2_{2,4}(\mathbf{x}'^{(2)} \otimes_{\mathrm{CG}} \mathbf{y}'^{(4)})^{(2)} + c^2_{4,2}(\mathbf{x}'^{(4)} \otimes_{\mathrm{CG}} \mathbf{y}'^{(2)})^{(2)} + c^2_{4,4}(\mathbf{x}'^{(4)} \otimes_{\mathrm{CG}} \mathbf{y}'^{(4)})^{(2)} \tag{45}$$

where the $c$'s are nonzero coefficients. Note that in order to have nonzero $(\mathbf{x}^{(2)} \otimes_{\mathrm{CG}} \mathbf{y}^{(2)})^{(2)}$ and $(\mathbf{x}^{(4)} \otimes_{\mathrm{CG}} \mathbf{y}^{(4)})^{(2)}$ contributions, $a_2, b_2, a_4, b_4$ must all be nonzero. However, that means we must have nonzero $(\mathbf{x}^{(2)} \otimes_{\mathrm{CG}} \mathbf{y}^{(4)})^{(2)}$ and $(\mathbf{x}^{(4)} \otimes_{\mathrm{CG}} \mathbf{y}^{(2)})^{(2)}$ contributions. Therefore weighted GTP is not expressive enough to output $(\mathbf{x}^{(2)} \otimes_{\mathrm{CG}} \mathbf{y}^{(2)})^{(2)} + (\mathbf{x}^{(4)} \otimes_{\mathrm{CG}} \mathbf{y}^{(4)})^{(2)}$, as it will necessarily mix additional terms.

## H DETAILS OF MESSAGE-PASSING NETWORK

In 2, we create learnable (ie, parametrized) variants of the purely functional tensor products. For the Clebsch-Gordan tensor product $\otimes_{\mathrm{CG}}$, we simply add a linear layer to its output. For the Gaunt

---

**Algorithm 2** LEARNABLETENSORPRODUCT

---

**Require:** Tensor Product $\otimes$, Number of Channels $C$ (for Gaunt tensor product).
  **procedure** LEARNABLETP($\mathbf{x}_1, \mathbf{x}_2$)
    **if** $\otimes = \otimes_{\mathrm{CG}}$ **then**
      **return** LINEAR($\mathbf{x}_1 \otimes_{\mathrm{CG}} \mathbf{x}_2$)
    **if** $\otimes = \otimes_{\mathrm{GTP}}$ **then**
      **for** $i = 1, 2, \ldots, C$ **do**
        $\mathbf{x}_1^{(i)} \leftarrow \text{LINEAR}_1^{(i)}(\mathbf{x}_1)$
        $\mathbf{x}_2^{(i)} \leftarrow \text{LINEAR}_2^{(i)}(\mathbf{x}_2)$
        $\mathbf{x}_o^{(i)} \leftarrow \text{LINEAR}_o^{(i)}(\mathbf{x}_1^{(i)} \otimes_{\mathrm{GTP}} \mathbf{x}_2^{(i)})$
      **return** CONCATENATE($\{\mathbf{x}_o^{(i)} \mid i \in \{1, 2, \ldots, C\}\}$)
  **return** LearnableTP

---

---

**Algorithm 3** Operation of our Message Passing Neural Network

---

**Require:** Graph $G$, Message Passing Iterations $T$, Cutoff $d_{\mathrm{max}}$, Spherical Harmonic Degree $\ell$, Tensor Product $\otimes$
  Compute neighbor lists for each node in $G$:

$$(u, v) \in E \iff \|\mathbf{r}_u - \mathbf{r}_v\| \leq d_{\mathrm{max}}$$

  Create LEARNABLETENSORPRODUCT from $\otimes$.
  **for** $v \in V$ **do**:
    $h_v^{(0)} \leftarrow [1]$
  **for** $t = 1, 2, \ldots, T$ **do**:
    **for** $v \in V$ **do**:
      $h_v^{(t)} \leftarrow \frac{1}{|\mathcal{N}(v)|} \sum_{u \in \mathcal{N}(v)} \text{MLP}(\|\mathbf{r}_u - \mathbf{r}_v\|) \times \text{LEARNABLETENSORPRODUCT}(h_u^{(t-1)}, Y_\ell(\mathbf{r}_u - \mathbf{r}_v))$
      $h_v^{(t)} \leftarrow \text{GATE}(h_v^{(t)})$
      $h_v^{(t)} \leftarrow \text{CONCATENATE}([h_v^{(t-1)}, h_v^{(t)}])$
      $h_v^{(t)} \leftarrow \text{LINEAR}(h_v^{(t)})$
  **return** $\{h_v^{(T)}\}_{v \in V}$

---

tensor product $\otimes_{\mathrm{GTP}}$, we create multiple channels, perform the tensor product channel-wise and then concatenate all irreps. This allows the output to have irreps of multiplicity $> 1$, even with the Gaunt tensor product. We set the number of channels $C$ as 4 in all experiments with the Gaunt tensor product.

In 3, we use these learnable tensor products in a simple message-passing network, very similar to NequIP (Batzner et al., 2022).

# I  ADDITIONAL BENCHMARKS

**Wall-Clock Time:** The elapsed time after compiling using `jax.jit`. To enable accurate measurements we calculate the mean wall-clock time for 100 rounds while performing 10 warmup rounds.

**Bandwidth and Throughput:** We used `Nsight Compute 2024.2.0.0 build 34181891` for profiling and reported Average GB/s and GFLOP/s from the individual kernel measurments using Roofline Hierarchical Analysis Yang (2020).

**GPU:** We gathered the GPU plots on an NVIDIA RTX A5500, running the CUDA driver version 550.90.07 and CUDA toolkit version 12.5. We use version 0.4.30 for `jax` and `jaxlib`.

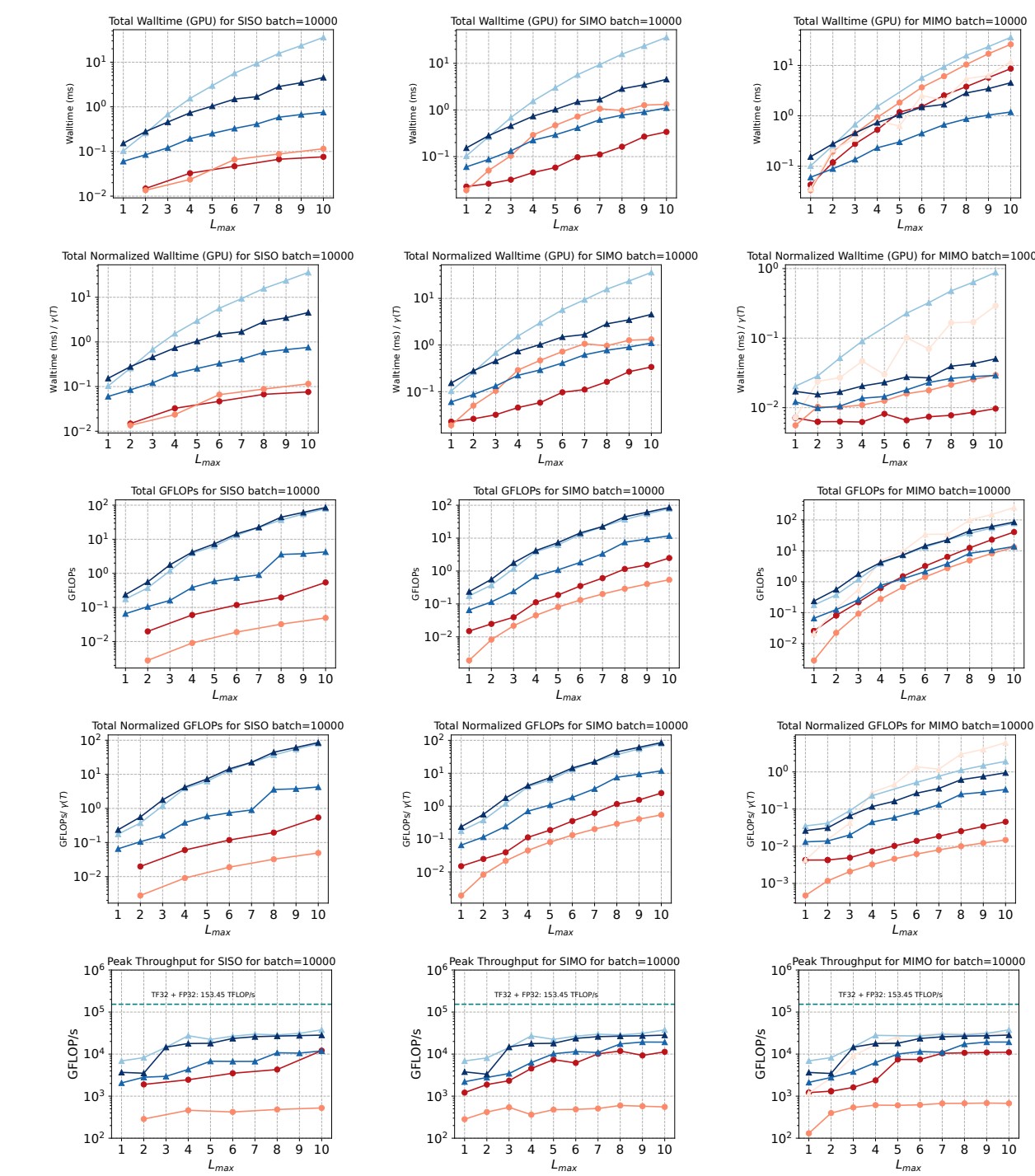

Figure 8: Analysis of SISO, SIMO and MIMO performance for different tensor products on RTX A5500 : Total walltime (top row), Total normalized walltime (second row), Total GFLOPs (third row) and Total normalized GFLOPs (bottom row). We had to skip some SISO $L_{max}$ values due to profiling errors.

## J PROOFS

### J.1 PROOF OF THEOREM 2.2

*Proof.* It is known that

$$Y_{\ell_1}^{m_1} \cdot Y_{\ell_2}^{m_2} \propto \sum_{j,\ell} C_{\ell_1,0,\ell_2,0}^{\ell,0} C_{j_1,m_1,j_2,m_2}^{j,m} Y_\ell^m$$

Gaunt (1929). Hence, we see that

$$(\mathbf{x}^{(\ell_1)} \otimes_{\text{GTP}} \mathbf{y}^{(\ell_2)})^{(\ell)} \propto C_{\ell_1,0,\ell_2,0}^{\ell,0} C_{j_1,m_1,j_2,m_2}^{j,m} (\mathbf{x}^{(\ell_1)} \otimes_{\text{CG}} \mathbf{y}^{(\ell_2)})^{(\ell)}.$$

For selection rule 1, this is inherited from the selection rules of CGTP. For selection rule 2, this follows from the selection rules for the $C_{\ell_1,0,\ell_2,0}^{\ell,0}$ term which is nonzero only when $\ell_1 + \ell_2 + \ell$ is even. $\qquad\square$

### J.2 PROOF OF THEOREM 3.5

*Proof.* It is known that

$$\mathbf{Y}_{j_1,\ell_1}^{m_1} \times \mathbf{Y}_{j_2,\ell_2}^{m_2} \propto \sum_{j,\ell} \begin{Bmatrix} j_1 & \ell_1 & 1 \\ j_2 & \ell_2 & 1 \\ j & \ell & 1 \end{Bmatrix} C_{\ell_1,0,\ell_2,0}^{\ell,0} C_{j_1,m_1,j_2,m_2}^{j,m} \mathbf{Y}_{j,\ell}^m$$

where $\begin{Bmatrix} j_1 & \ell_1 & 1 \\ j_2 & \ell_2 & 1 \\ j & \ell & 1 \end{Bmatrix}$ is a Wigner 9j symbol (James, 1976; Varshalovich et al., 1988). Hence, we see that

$$(\mathbf{x}^{(j_1,\ell_1)} \otimes_{\text{VSTP}} \mathbf{y}^{(j_2,\ell_2)})^{(j,\ell)} \propto \begin{Bmatrix} j_1 & \ell_1 & 1 \\ j_2 & \ell_2 & 1 \\ j & \ell & 1 \end{Bmatrix} C_{\ell_1,0,\ell_2,0}^{\ell,0} (\mathbf{x}^{(j_1,\ell_1)} \otimes_{\text{CG}} \mathbf{y}^{(j_2,\ell_2)})^{(j,\ell)}.$$

For selection rule 1, we note that it follows from Definition 3.1. For selection rule 2, this is inherited from the selection rules of CGTP. For selection rules 3 and 4, this follows from the selection rules for the $C_{\ell_1,0,\ell_2,0}^{\ell,0}$ term.

For selection rule 5, we use the symmetry properties of the Wigner 9j symbols. Suppose there exists a permutation $a, b, c$ such that $j_a = \ell_a$ and $(j_b, \ell_b) = (j_c, \ell_c)$. Suppose we swap rows $b, c$, this is an odd permutation so the symmetries of the 9j symbol means we pick up a phase factor $(-1)^S$ where $S = \sum_{i=1}^3 (j_i + \ell_i + 1)$. Note that $S$ is odd because each $j_b + \ell_b = j_c + \ell_c$ and $j_a = \ell_a$. Hence our phase factor is $-1$. But swapping $b, c$ does not change the 9j symbol since $j_b = j_c$. Hence by symmetry the 9j symbol must vanish, giving us selection rule 5. $\qquad\square$

### J.3 PROOF OF THEOREM 3.6

*Proof.* We already satisfy condition 2. Without loss of generality, assume $j_1 \leq j_2 \leq j_3$

*Case 1:* Suppose $j_1, j_2, j_3$ are distinct. Condition 5 is already satisfied since the $j$'s are unique. Since the $j$'s are distinct integers, we have $j_1 + 1 \leq j_2 \leq j_3 - 1$. If $j_1 + j_2 + j_3$ is even, we can set $\ell_i = j_i$ so conditions 1, 3, 4 are clearly satisfied. If $j_1 + j_2 + j_3$ is odd, we can set $\ell_1 = j_1, \ell_2 = j_2, \ell_3 = j - 3$. By construction 1, 4 are satisfied. For 3, we have

$$\ell_3 < j_3 \leq j_1 + j_2 = \ell_1 + \ell_2$$
$$\ell_2 = j_2 \leq j_3 - 1 = \ell_3 \leq \ell_3 + \ell_1$$
$$\ell_1 = j_1 \leq j_3 - 1 = \ell_3 \leq \ell_3 + \ell_2.$$

Hence we can always choose $\ell_1, \ell_2, \ell_3$ which satisfy the selection rules.

*Case 2:* Suppose two of $j_1, j_2, j_3$ are equal. Then we have two subcases.

*Subcase 1:* $j_1 = j_2 \leq j_3 - 1$. Note that $j_3 - 1 \geq 0$ so $j_3 \geq 1$. But $1 \leq j_3 \leq j_1 + j_2 = 2j_1$ so $j_1 \geq 1$ since $j_1$ is an integer.

If $j_1 + j_2 + j_3$ is even, we can set $\ell_1 = j_1, \ell_2 = j_2 + 1$, and $\ell_3 = j_3 - 1$. By construction we satisfy $1, 4$. Since $\ell_1 \neq \ell_2$ and $j_1 = j_2 \neq j_3$ we will satisfy $5$. For $3$ we find that

$$\ell_1 = j_1 = j_2 < j_2 + 1 = \ell_2 \leq \ell_2 + \ell_3$$

$$\ell_2 = j_2 + 1 \leq j_3 = \ell_3 + 1 \leq \ell_1 + \ell_3$$

$$\ell_3 = j_3 - 1 < j_1 + j_2 = \ell_1 + \ell_2 - 1 < \ell_1 + \ell_2.$$

Hence we satisfy all the selection rules.

Suppose $j_1 + j_2 + j_3$ is odd. Then we can set $\ell_1 = j_1, \ell_2 = j_2 + 1$, and $\ell_3 = j_3$. By construction we satisfy $1, 4$. Since $\ell_1 \neq \ell_2$ and $j_1 = j_2 \neq j_3$ we will satisfy $5$. For $3$ we find that

$$\ell_1 = j_1 = j_2 < j_2 + 1 = \ell_2 \leq \ell_2 + \ell_3$$

$$\ell_2 = j_2 + 1 \leq j_3 = \ell_3 \leq \ell_1 + \ell_3$$

$$\ell_3 = j_3 \leq j_1 + j_2 = \ell_1 + \ell_2 - 1 < \ell_1 + \ell_2.$$

Hence we satisfy all the selection rules.

*Subcase 2:* $j_1 + 1 \leq j_2 = j_3$. If $j_1 + j_2 + j_3$ is even, we can set $\ell_1 = j_1 + 1$, $\ell_2 = j_2$, and $\ell_3 = j_3 - 1$. By construction we satisfy $1, 4$. Since $\ell_2 \neq \ell_3$ and $j_2 = j_3 \neq j_1$, we satisfy $5$. For $3$ we find

$$\ell_1 = j_1 + 1 \leq j_2 = \ell_2 \leq \ell_2 + \ell_3$$

$$\ell_2 = j_2 \leq j_1 + j_3 = (j_1 + 1) + (j_3 - 1) = \ell_1 + \ell_3$$

$$\ell_3 = j_3 - 1 \leq j_1 + j_2 - 1 < j_1 + 1 + j_2 = \ell_1 + \ell_2.$$

Hence we satisfy all the selection rules.

If $j_1 + j_2 + j_3$ is odd, we can set $\ell_1 = j_1 + 1$, $\ell_2 = j_2$, and $\ell_3 = j_3$. By construction we satisfy $1, 4$. Since $j_1 \neq \ell_1$ and $j_2 = j_3 \neq j_1$, we satisfy $5$. For $3$ we find

$$\ell_1 = j_1 + 1 \leq j_3 = \ell_3 \leq \ell_2 + \ell_3$$

$$\ell_2 = j_2 \leq j_1 + j_3 < (j_1 + 1) + j_3 = \ell_1 + \ell_3$$

$$\ell_3 = j_3 \leq j_1 + j_2 < (j_1 + 1) + j_2 = \ell_1 + \ell_2.$$

Hence we satisfy all the selection rules.

*Case 3:* Suppose $j_1 = j_2 = j_3 = j$. If $j > 0$ and is even, then $j \geq 2$. We can pick $\ell_1 = j - 1, \ell_2 = j, \ell_3 = j + 1$. By construction we satisfy $1, 4$. Since the $\ell$'s are distinct we also satisfy $5$. For $3$ we check that

$$\ell_1 = j - 1 < j + j + 1 = \ell_2 + \ell_3$$

$$\ell_2 = j < j - 1 + j + 1 = \ell_1 + \ell_3$$

$$\ell_3 = j + 1 = j - 1 + 2 \leq j - 1 + j = \ell_1 + \ell_2.$$

Hence we satisfy all the selection rules.

If $j$ is odd then $j \geq 1$. We can pick $\ell_1 = j - 1, \ell_2 = j, \ell_3 = j$. By construction we satisfy $1, 4$. Since $\ell_1 \neq j$ and $\ell_2 = \ell_3 \neq \ell_1$ we also satisfy $5$. For $3$ we check that

$$\ell_1 = j - 1 < j + j = \ell_2 + \ell_3$$

$$\ell_2 = j \leq j - 1 + j = \ell_1 + \ell_3$$

$$\ell_3 = j \leq j - 1 + j = \ell_1 + \ell_2.$$

Hence we satisfy all the selection rules.

The only case which fails is $j_1 = j_2 = j_3 = 0$ in which case selection rule $1$ forces $\ell_1 = \ell_2 = \ell_3 = 1$ which breaks rule $4$. However, this case just correspond to multiplication of scalars which is trivial. $\qquad\square$