# OpenReview forum: "The Price of Freedom: Exploring Tradeoffs in Equivariant Tensor Products with Spherical Signals"
_ICLR.cc/2025/Conference — ICLR 2025 Conference Withdrawn Submission_

### Official Review · Reviewer_bNV6 · 2024-11-01

**Soundness:** 3
**Presentation:** 2
**Contribution:** 2
**Rating:** 3
**Confidence:** 3

**Summary:**

The paper investigates tensor products in E(3)-equivariant neural networks, which are pivotal for 3D modeling tasks. The authors analyzed existing optimization techniques and proposed new variants. To overcome the issue that GTP cannot perform antisymmetric tensor products,
they introduce the Vector Signal Tensor Product (VSTP) as a solution that allows antisymmetry and generalizes it to a class of irrep signal
tensor products. Finally, the authors provide a comparative analysis of runtime performance and expressivity across various tensor products.

**Strengths:**

1. The authors comprehensively reviewed and investigated existing techniques to optimize the tensor products.
2. A new technique named VSTP was proposed based on GTP to overcome the issue that GTA cannot compute antisymmetric tensor products.
3. The tradeoff between computation efficiency and expressivity is thoroughly discussed in the paper.

**Weaknesses:**

1. The paper is very technical and is hard to follow for a person who is not an expert in this area. It might be better to include more background and introductions about related problems, concepts, and notations.
2. The organization of the paper can be further improved. Section 2 spends lots of space discussing existing techniques, but CGTP and MTP are only used for comparisons in experiments. It would be better if the authors could include more novel contents in the main body of the paper and move the review to the appendix. For instance, more details can be included in Section 3 and Section 4, which appear to be the novelty and most interesting part of the paper.
3. The title does not correctly reflect the content of the paper. In my understanding, only a small portion of Section 2 and Section 4 talk about the tradeoff. Lots of contents of the paper are not very related to the title.

**Questions:**

1. In line 195 and line 229, authors mentioned GTP and MTP both lose information. Could authors add more explanations about why and how information is lost in these two methods?
2. In line 401, the authors said that the only true asymptotic speedup comes from S2FFT algorithm. Could authors add more details about this algorithm and explain why it differs from other methods like GTP and MTP.
3. For the experiment in Section 5.1, Clebsch-Gordan (Sparse) having the lowest FLOPs yet a high GPU walltime. What is the reason causing this weird phenomenon?
4. For Section 5.2, could the authors also include training time of CGTP and GTP? Can the proposed VGTP solve the problem in Section 5.2 and get speedup?

---

### Official Review · Reviewer_awUu · 2024-11-04

**Soundness:** 3
**Presentation:** 3
**Contribution:** 2
**Rating:** 3
**Confidence:** 4

**Summary:**

The paper focuses on speeding up tensor products of irreps in E(3)-equivariant neural networks. The authors introduce the vector signal tensor product (VSTP), and its extension called irrep signal tensor products (ISTPs). The paper provides a theoretical analysis of expressivity and runtime.

**Strengths:**

The structure of the paper is clear. It presents complex concepts in a way that is easier to follow for readers familiar with equivariant neural networks. The clear structure helps in understanding the theoretical foundations, proposed methods, and experimental evaluations.

**Weaknesses:**

Major points:

- The main difference of VSTP over existing works is its support for antisymmetric operations. However, the significance of considering antisymmetry for performance enhancement in real-world settings remains unclear. The single synthetic experiment in Section 5.2 does not convincingly establish that antisymmetry is critical in real-world tasks, leaving the practical importance of the contribution somewhat uncertain.
- The authors did not actually apply VSTP to a model on relevant tasks (e.g. molecular modeling) to demonstrate actual improvements, leaving questions about its practical applicability.

Minor point:

The word 'section' is duplicated at the end of Line 111.

**Questions:**

- Could the authors provide examples of several well-known real-world tasks where accounting for antisymmetry significantly enhances performance without introducing additional overhead?
- Could the authors actually implement the VSTP in models and show its improvements in practical applications?

---

### Official Review · Reviewer_9XVG · 2024-11-04

**Soundness:** 2
**Presentation:** 2
**Contribution:** 2
**Rating:** 5
**Confidence:** 3

**Summary:**

The paper addresses a significant challenge in the development of E(3)-equivariant neural networks by examining different tensor product operations and introducing new methods (VSTP and ISTPs) to enhance both efficiency and expressivity. Through theoretical analysis and empirical benchmarks, the authors provide a detailed comparison of existing tensor products, highlighting their respective limitations in runtime and expressivity.

**Strengths:**

1. The paper is well-structured.

2. By introducing the Vector Signal Tensor Product (VSTP) and generalizing to Irrep Signal Tensor Products (ISTPs), the authors address a notable limitation in existing models—namely, the inability to handle antisymmetric operations—thereby enhancing the expressivity of equivariant neural networks.

3. Theoretical foundations are robust, with well-defined selection rules for various tensor products, contributing valuable mathematical insights that enhance the paper's technical rigor.

4. The paper provides valuable insights by analyzing the asymptotic runtimes and expressivity of various tensor product implementations.

**Weaknesses:**

1. The experimental results for the proposed methods, VSTP and ISTPs, are insufficient, making the contributions of the proposed methodology unclear. I would like to see the methods applied to models such as eSCN, MACE, and EquiformerV2, with comparisons to GTP

2. It is difficult to follow the authors' arguments and understand their main claims. Several mathematical notations for tensor products are not provided. For example, in Equation 2, terms like l_1, l_2, and "tosphere" are introduced without explanation, as well as "Vector Gaunt" in Figure 6.

3. Certain mathematical sections, especially in the appendices, are densely packed and could be challenging for readers who are not deeply familiar with representation theory, potentially limiting accessibility.

4. The paper does not provide a comparative analysis with alternative architectures, focusing narrowly on specific methodologies without contrasting them to other advanced approaches, which could offer a more comprehensive perspective on the proposed methods.

**Questions:**

See weaknesses.

---

### Official Review · Reviewer_9meG · 2024-11-10

**Soundness:** 2
**Presentation:** 2
**Contribution:** 3
**Rating:** 5
**Confidence:** 4

**Summary:**

This paper looks at some of the current tensor product operations proposed for E(3)-equivariant neural networks: the Clebsch-Gordan tensor product, the Gaunt tensor product, and the matrix tensor product. The authors analyze runtimes (using wall time and FLOP count and theoretical runtimes) of the different tensor products. They also introduce some new tensor product operations such as sparse CG, ISTP, and VSTP.

**Strengths:**

- The asymptotic runtimes of the different tensor products is interesting, and many different tensor product implementations are included
- The additional improvements on current tensor product implementations could open up studying new improvements for the tensor product operations in equivariant neural networks

**Weaknesses:**

- The authors define a new measure, B, and the dimension of this measure as a proxy for expressivity of the tensor product. However, this choice does not seem well justified, especially in the context of using this tensor product with neural networks. Neural network expressivity is affected by much more than this.

- The authors say in Section 5 that the Clebsch-Gordan tensor products are the fastest both in terms of wall time and FLOPs after normalizing by \gamma(T). What does this mean? This statement seems far too strong to make, when it is not clear if it makes sense to normalize by \gamma(T) or if this is even a sound measure.

- The conclusions the authors are coming to are too strong for limited evaluation: for example the wall time comparisons seem to only be run on an RTX A5500.

- The FLOP count reported here may be an “idealized” FLOP count if the authors used the Nsight profiler. Thus, these numbers are likely to not be accurate.

- The practical performance of these different tensor products seems to also depend on how well optimized the different tensor products are. More discussion of this would be helpful, as otherwise it may not be a true one to one comparison.

- There is only one example in this paper, the Tetris example. It is not clear what the practical utility of the examples in this paper is. More use cases and examples would be helpful, such as substituting the new tensor product implementations into current models to show performance or speed improvements. For example, it seems like other papers like the Gaunt Tensor Product paper implemented the tensor product into other neural networks on datasets such as OC20 and 3BPA. Or what about looking at molecular datasets that have chiral molecules?

**Questions:**

- The expressivity measure is poorly defined and justified, especially with how this relates to the expressivity of a neural network using these tensor products. Can you justify this more?

- “Normalizing” for expressivity also is poorly defined. Can you please explain this in more detail?

- It does not make sense to me why the last column in Table 1 is grouped as “Runtime / Expressivity.” What does this mean?

- What about substituting the different tensor product implementations into current models to assess performance or speed improvement? This could include assessing the performance on datasets such as OC20 or other molecular datasets.

- The practical performance and wall time seems like it may depend a lot on hardware and optimization of implementation. Is there more analysis here that can be done, rather than coming to conclusions only based off of wall times on a RTX A5500?

---

### Note · Authors · 2024-11-27

**Comment:**

We would like to thank all the reviewers for their time in reading our work and giving detailed feedback. One common theme across all reviews is the focus on VSTP/ISTP as our main result. We realize that our original submission seems to suggest that this is the focus which is not the case in reality. Instead, our main point is our comprehensive analysis of the differences between tensor products and that it provides a framework for analyzing new proposed tensor products. We apologize for the lack of clarity in our original submission. After much discussion, we decided we need to make significant changes to reframe our work, including removing VSTP/ISTP which only distracts from the main message. Instead, we decided to expand more on the discussion of how to properly analyze and benchmark existing tensor products and submit VSTP/ISTP as a separate smaller work.

However, we realize that it is unfair for the reviewers to reevaluate such significant changes, hence we decide to withdraw our submission. We would once again like to thank the reviewers for their feedback which will be extremely valuable for improving the clarity and framing of our future submissions.

**Withdrawal Confirmation:**

I have read and agree with the venue's withdrawal policy on behalf of myself and my co-authors.